# A Theoretical design of Concept Sets: improving the predictability of concept bottleneck models

**Max Ruiz Luyten**
University of Cambridge
Cambridge CB3 0WA
mr971@cam.ac.uk

**Mihaela van der Schaar**
University of Cambridge
Cambridge CB3 0WA
mv472@cam.ac.uk

## Abstract

Concept-based learning, a promising approach in machine learning, emphasizes the value of high-level representations called concepts. However, despite growing interest in concept-bottleneck models (CBMs), there is a lack of clear understanding regarding the properties of concept sets and their impact on model performance. In this work, we define concepts within the machine learning context, highlighting their core properties: *expressiveness* and *model-aware inductive bias*, and we make explicit the underlying assumption of CBMs. We establish theoretical results for concept-bottleneck models (CBMs), revealing how these properties guide the design of concept sets that optimize model performance. Specifically, we demonstrate that well-chosen concept sets can improve sample efficiency and out-of-distribution robustness in the appropriate regimes. Based on these insights, we propose a method to effectively identify informative and non-redundant concepts. We validate our approach with experiments on CIFAR-10 and MetaShift, showing that concept-bottleneck models outperform the foundational embedding counterpart, particularly in low-data regimes and under distribution shifts. We also examine failure modes and discuss how they can be tackled.

## 1 Introduction

Concept Bottleneck Models (CBMs) [23] offer a compelling trifecta of interpretability [19], intervenability, and predictability [28]. They are inspired by the power of concepts: the fundamental building blocks of human thought that guide our learning, decision-making, inference, and communication [13]. In a nutshell, CBMs force these high-level representations to inform model predictions. For example, rather than directly predicting a car's action from raw images, a CBM first identifies key concepts such as the presence of a semaphore or a pedestrian crossing the road ($C : \mathcal{X} \to \mathcal{C}$), and learns $\hat{g} : \mathcal{C} \to \mathcal{Y}$ to act based on them ($(\hat{g} \circ C)(x)$). Indeed, despite never having seen a Tesla Model X, one could reasonably classify them provided they can identify an SUV, a T logo, gull-wing doors, and the absence of exhaust pipes. This paradigm has found applications across various domains, including automated driving [12], fine segmentation [31], and clinical imaging [39, 22].

Despite their promise, the predictability of CBMs, particularly in relation to the properties of the concept set $\mathcal{C}$, remains poorly understood. A deeper understanding of how the composition and characteristics of concept sets affect CBM performance is essential to enhance their practical utility and robustness in real-world applications.

In this paper, we address this gap by systematically investigating the properties of concept sets $\mathcal{C} = \{c_1, \ldots, c_m\}$ and their influence on model performance. We answer the following key questions:

38th Conference on Neural Information Processing Systems (NeurIPS 2024).

1. **How do characteristics of the concept set impact the performance of the CBM?** We identify the size of the concept set $|\mathcal{C}|$ and the degree of misspecification $\varepsilon$ as primary drivers of the performance of the CBM, where $\varepsilon := ||\arg\min_{f \in \mathcal{H}} \ell(f) - \arg\min_{g \in \mathcal{H}_\mathcal{C}} \ell(g \circ C)||$, with $\mathcal{H} \subseteq \{f : \mathcal{X} \to \mathcal{C}\}, \mathcal{H}_\mathcal{C} \subseteq \{g : \mathcal{C} \to \mathcal{Y}\}$ the hypotheses spaces.

2. **What are the optimal conditions for the effectiveness of CBMs?** We find that CBMs are particularly effective in small data regimes and under distribution shifts.

To address these questions, we introduce in Section 3 a theoretical framework that links the properties of concept sets to CBM performance metrics. This framework identifies two core *desiderata* for concept sets: *expressiveness* and *model-aware inductive bias*. *Expressiveness* ensures each concept captures valuable information from inputs, benefiting the expected loss $\mathcal{R}(\hat{g} \circ C)$ as the sample size $n \to \infty$. *Model-aware inductive bias* implies a beneficial inductive bias, improving performance in low-sample regimes ($n \approx 0$). These properties highlight an implicit assumption in CBMs about the alignment between model outputs and human inductive biases.

In Section 4, we dive into a theoretical analysis of CBMs. Our setting aligns with the recent trend of concept bottleneck models built on multimodal foundational models such as CLIP [30, 39, 41]. We derive a key theorem (Theorem 1) that compares the expected loss of concept-based models ($\mathcal{R}(\hat{g} \circ C), \hat{g} \in \arg\min_{g \in \mathcal{H}_\mathcal{C}} \ell(g \circ C)$) with the baseline counterpart ($\mathcal{R}(\hat{f}), \hat{f} \in \arg\min_{f \in \mathcal{H}} \ell(f)$). The theorem demonstrates that concept-based models can outperform baseline models if the concept set is well chosen by minimizing the size of the concept set $|\mathcal{C}|$ while controlling the misspecification term $\varepsilon$. This theoretical insight highlights the importance of selecting informative and non-redundant concepts, as proper concept selection can significantly affect sample efficiency and generalization performance.

Based on these insights, we propose a method in Section 5 to generate informative and non-redundant concept sets. We also discuss potential failure modes of CBMs and how existing approaches can mitigate them. Finally, we empirically validate our framework in Section 6 with experiments on CIFAR-10 and MetaShift datasets, showing that appropriately designed CBMs outperform traditional models, particularly in low data and distribution shift scenarios.

**Contributions.**

1. **Theoretical Framework for Concept Sets**: We introduce a novel theoretical framework for CBMs that elucidates the properties of concept sets, specifically focusing on *expressiveness* and *model-aware inductive bias*. This framework formalizes how these properties influence model performance, offering new insights into the design of effective concept sets.

2. **Theoretical Analysis and Insights**: We derive theoretical results that compare the expected loss of CBMs with their non-bottleneck counterparts. Our analysis reveals the conditions under which CBMs are most effective, particularly in low-data regimes and under distribution shifts. These results provide a clear understanding of the design choices that enhance CBM performance.

3. **Empirical validation**: We validate our results through experiments on CIFAR-10 and MetaShift datasets, designed to assess different aspects of concept sets and CBMs, and we affirm that appropriately designed concept sets can lead to increased model predictability and efficiency.

## 2   Related Work

Incorporation of human concepts into machine learning algorithms has been considered for a long time as a strategy to improve the explainability of the model through improved representation interpretability [21, 27, 20, 38]. Concept Bottleneck Models (CBMs) [23] are one of the most promising approaches to concept-based methods and have been adopted in various applications such as fine classification [31] and medical image analysis [39, 22].

**Advances in concept identification**    Recent work has focused on increasing the flexibility of CBMs. For example, [30, 41, 40] explore the use of multimodal and large language models for concept identification to eliminate the need for predefined labels. These methods enhance the adaptability of CBMs to various tasks and domains, improving their practical utility.

**Performance Enhancement Strategies**    Recent works consider relaxing the bottleneck by post hoc concept discernment [41] or fitting the CBMs' residuals [36] to improve model accuracy while preserving the other properties of CBMs.

**Gaps and Our Contribution**    Despite the mixed successes of these efforts, there remains a gap in understanding concept bottleneck models and, in particular, how the properties of concept sets affect their performance.

In summary, our contribution is orthogonal to previous work, providing a deeper theoretical understanding of the relationship between concept set properties and model performance.

## 3    What are concepts?

Although the notion of concept has been repeatedly used in the ML literature [23, 30, 10], there is still a lack of a precise definition and delineation of their properties. To make them theoretically tractable entities, we begin by addressing what concepts are.

According to cognition theory, concepts are mental representations that constitute a fundamental key building block in all aspects of cognition [15]. More specifically, they allow us to draw appropriate inferences about the type of entities we encounter in our everyday lives and are necessary for cognitive processes such as categorization, memory, decision-making, learning, and inference. Thus, the formalism should reflect that (1) *concepts are representations* and (2) *concepts empower inferences for everyday tasks*.

**Concepts are representations**    Let's denote a hypothetical set of concepts as $\mathcal{C}$ and a particular concept as $c$. As representations of some inputs $\mathcal{X}$, concepts can be regarded as mappings from this space. To define the range of these mappings, we emphasize that concept representations indicate 'membership' in a class. In modern theories (e.g., Prototype Theory), concepts consist of fuzzy memberships instead of binary ones. For example [16], humans consider "sink" as borderline a "kitchen utensil", or "octopuses" are just excluded from the concept of "fish". As such, concepts lend themselves to the fuzzy mathematical formulation of membership functions, a generalization of characteristic functions. Thus, the concept can be unequivocally represented as a mapping $c : \mathcal{X} \to [0, 1]$.

**Concepts aid in everyday-task inference**    Not all membership functions qualify as concepts; the missing component is their usefulness in task-specific inference. For a new concept to be valuable, it must improve predictability for a subset of tasks relevant to humans. This can happen either by capturing new nuances of inputs (for example, new color tones) or integrating different concepts into a more comprehensive entity (e.g., grouping animals into families).

Consider a task $\mathcal{T}$, defined as a tuple[1] $(\mathcal{D}, \ell)$, where $\mathcal{D} = \{(x_i, y_i) \sim_{iid} \mathbb{P}(\mathcal{X}, \mathcal{Y})\}$ is a dataset drawn from the data generation process $\mathbb{P}(\mathcal{X}, \mathcal{Y})$ and $\ell$ is a loss function. Define the $n$-risk of a model class $f_\Theta$ on a task $\mathcal{T}$ as the expected loss of a model trained on $n$ samples, $\mathcal{R}_n(f_\Theta) = \mathbb{E}_{\mathcal{D}, x, y} \left[ \ell(\hat{f}(x; \theta(\mathcal{D})), y) \right]$. Consider the output hypothesis spaces $\mathcal{H}_\mathcal{C} \subseteq \{g : \mathcal{C} \to \mathcal{Y}\}$ complete under permutation symmetry and the learning algorithm invariant under feature permutations. We refer to Appendix B.1 for a technical discussion of this assumption.

---

[1]We assume static tasks for simplicity, but we could admit the superclass of multi-step tasks, arguably more typical for humans. Also, for simplicity, we consider the effect of a new concept on a single task. Still, one might want to consider the distribution of human tasks and compute the expected improvement in risk across all tasks.

**Definition 1.** *[Concept set] Let $\hat{f}(\cdot, \theta(\mathcal{D})) : \mathcal{X} \to \mathcal{Y}$ be a model trained on the task $\mathcal{T} = (\mathcal{D}, \ell)$, and $\mathcal{C}$ a preexisting concept set, potentially void, of size $m - 1$. Given a membership function $c_m$, we denote the conceptualized model $\hat{f}_{\mathcal{C} \cup \{c_m\}} = \hat{g}_m(\cdot, \theta_c) \circ [c_1, \ldots, c_m]$, such that $\hat{g}_m : \mathcal{C} \cup \{c_m\} \to \mathcal{Y}$ is trained on the conceptualized input $\{(c_j(x_i))_{j \in [m]}\}_{i \in [n]}$.*

*Then, $\mathcal{C} \cup \{c_m\}$ is a concept set if there exists $N_0 \leq N_1 \in \mathbb{N}$ such that*

$$\mathcal{R}_n(\hat{f}_{\mathcal{C}}) > \mathcal{R}_n(\hat{f}_{\mathcal{C} \cup \{c_m\}}) \quad \text{for } N_0 \leq n \quad \text{(expressiveness)}$$
$$\mathcal{R}_n(\hat{f}) > \mathcal{R}_n(\hat{f}_{\mathcal{C} \cup \{c_m\}}) \quad \text{for } n \leq N_1 \quad \text{(model-aware inductive-bias)}$$

This implies that the output model trained on the concept set generalizes better than the vanilla one in the low-sample domain (model-aware inductive bias) and better than the smaller concept set in the large-sample domain (expressiveness)[2]. We further examine the role of these properties in the Appendix B.2.

Note that for human concept sets, the hypothesis space for the learned functions $\hat{f}, \hat{g}_m$ corresponds to the human model. This definition can be regarded not only as a qualifying condition for a membership function to be considered a concept, but also as a measure of its quality, e.g., through the ratio of risks. From this we could discover iteratively new concepts by greedily maximizing the decrease in risk for a new concept or even take a hierarchical approach by unveiling the most relevant concepts to identify a given set of already established concepts.

**The CBM Assumption**  An implicit assumption behind concept-based models that we make explicit is that such human concepts will also have predictive power for a given ML output model. The output model is conventionally linear for interpretability.

Although expressiveness and model-aware inductive bias may not always transfer across different models, it is reasonable to assume that a representation providing valuable inductive bias for one model can be beneficial for others. Additionally, given ML tasks are by design relevant to humans, and on many ML tasks, humans show a non-trivial performance (e.g., image classification), it is reasonable to expect that human *Concept Sets* can benefit ML models too in these cases. However, an empirical validation (section 6) is required to check that the CBM assumption holds.

Furthermore, since humans have been exposed to a highly varied distribution, concepts are unlikely to be entangled with spurious features, which we also test in our insight experiments.

**Model agnostic concepts**  In some cases, it may be useful to evaluate the quality of a concept set without knowing the output model. We propose using an information-theory metric: $D_{\mathrm{KL}}[\mathbb{P}(Y|\mathcal{C} \cup \{c_m\}(X)) \| \mathbb{P}(Y|\mathcal{C}(X))] \gg 0$. A new concept should provide significant information on the output.

In the subsequent sections, we will dive into the theoretical insights derived from this formalism and explore empirical validations through various experiments.

**Natural benefits of concepts**  Using a concept representation rather than a typical pre-trained model representation provides direct benefits. One notable advantage is that human concepts allow us to decouple the representation model from the output model. This flexibility means that we can replace the underlying concept identification model with a more advanced version in the future while keeping the output model unchanged. The concept set serves as a stable communication interface with fixed dimensionality and functionality. In addition, concepts can act as common representations across different feature sets, potentially harmonizing datasets and modalities. We expand on this discussion in the appendix B.3.

---

[2]Interestingly, if we accept these properties for concepts, we could investigate what the ML models that most align with the human predictive model are, and thus provide plausible hypotheses for the potential architecture of the human brain.

# 4 Theoretical results

We now dive into the effect of concept-based representations and their theoretical advantages. This includes the potential for improved data efficiency, as the concept-based representation space is inherently more structured and informative, thereby reducing the amount of data required to achieve comparable performance. Additionally, we will delve into the mechanisms by which concept-based representations enhance model robustness to distribution shifts. In the process, it will also become apparent in what regimes it is undesirable to bottleneck a model on concepts and potential solutions.

In our analysis, we focus on a recently popular approach to concept-based models, that is, concept bottleneck models based on multi-modal foundational models, such as CLIP [30, 39, 41].

To that end, we consider the input space $\bar{\mathcal{X}}$ as the joint embedding space of inputs and concepts from a foundational model, where it is assumed [21, 41, 6] that concepts can be related to directions, and it is typical to use a linear output layer on top of such representation.

**Definition 2** (Risk). *Let* $x, x^{(i)} \overset{i.i.d.}{\sim} \mathbb{P}(\mathcal{X})$ *for* $i \in [n]$, *and let* $y^{(i)} = f^*(x^{(i)})$. *Given a loss* $\mathcal{L} : \mathcal{Y} \times \mathcal{Y} \to \mathbb{R}$, *the risk of a predictor* $\hat{f}$ *trained on the* $n$ *samples* $(X, y)$ *is given by*

$$\mathcal{R}(\hat{f}) = \mathbb{E}_{x,X} \left\{ \mathcal{L} \left[ f^*(x), \hat{f}(x) \right] \right\} . \tag{1}$$

Now, since the output layer is conventionally linear, we may write $f^*(x) = \omega^* x$ and $\hat{f}(x) = \hat{\omega} x$. We further consider the $l_2$ loss, where the predictor takes the analytical form $\hat{\omega} = Y X^\dagger$, $X^\dagger$ representing the Moore-Penrose pseudoinverse of $X$.

Given concepts are represented by directions in this space, we can write a set of concepts $\mathcal{C}$ as a set of dual vectors $\{c^{(1)}, \ldots, c^{(k)}\} \in \bar{\mathcal{X}}^*$, we can consequently consider our conceptual representation as $CX$ with $C = [c^{(1)}, \ldots, c^{(k)}]^\mathsf{T}$, and our overall concept model as $\hat{f}_C = \hat{g}_c \circ C$.

We now turn to the **Model-aware inductive bias** condition and assess whether there are more explicit conditions that imply it in practice, and thus compute the risk for the concept embedding model and the baseline.

**Theorem 1.** *Under the above setting and assuming that* $\mathbb{P}$ *is an isotropic distribution on the* $d-$*dimensional input, then the risk of the concept-based model* $\mathcal{R}(\hat{f}_C)$ *is given by*

$$\mathcal{R}(\hat{f}_C) = \left(1 - n\frac{d - |\mathcal{C}|}{(d+2)(d-1)}\right) \mathcal{R}(\hat{f}_b) + n \left(\frac{1}{d} + \frac{d-n}{(d+2)(d-1)}\right) \varepsilon,$$

*where* $\varepsilon = ||\omega^* C^\perp|||_F^2$, *and* $C^\perp$ *stands the orthogonal projection operator onto the orthogonal complement of the concept vector's span.*

We note that this result aligns with our intuition. Indeed, even without spurious features or distribution shifts, concepts yield a practical advantage over the baseline foundational embedding (first term), but only if concepts are adequately chosen (second term). Let's dive deeper into each of these two terms.

Let us start with the $\varepsilon$ term that corresponds to the misspecification of the concept set. Certainly, the larger the component of the actual predictor that does not belong to the subspace captured by the concepts, the lesser the quality of our concept model. Note how the $\varepsilon$ coefficient grows with the number of examples $n$, which means that the misspecification term is more influential when more examples are available. Nonetheless, with a proper choice of the concept set, the $\varepsilon$ can be nullified.

When $\varepsilon \approx 0$, the remaining terms directly compare the concept and the baseline risks $\mathcal{R}(\hat{f}_C), \mathcal{R}(\hat{f}_b)$. In fact, in this situation, we have a strict benefit for using the concept model, that is, $\mathcal{R}(\hat{f}_C) < \mathcal{R}(\hat{f}_b)$. Interestingly, provided $\varepsilon \approx 0$, the remaining term suggests we should aim for the minimal set of concepts $|\mathcal{C}|$, and we obtain a better scaling with the number of examples (i.e., increased sample efficiency).

Now, let us focus on the setting of spurious features due to confounders. As a motivating example, imagine the image classification setting, in which the classes are confounded with an environment property (e.g., outdoor dogs vs indoor cats). In this case, the image setting acts as a confounder $Z$, manifesting as a spurious feature in the images. We note that such confounders are human concepts, which, per our previous discussion, are embodied by directions in the CLIP embedding $z \in \bar{\mathcal{X}}^*$. Then,

**Theorem 2.** *In the presence of confounders $\{z_1, z_2, \ldots, z_k\} \in \bar{\mathcal{X}}^*$, given a concept $c \in \bar{\mathcal{X}}^*$, the bias in the ordinary least squares estimator satisfies*

$$||Bias(\hat{f}_C|X)||_2 \propto ||(\langle c, z_1 \rangle, \ldots, (\langle c, z_k \rangle))||_2$$

*In particular, the classifier $\hat{f}_{\{c\}}$ remains unbiased if and only if $c \in span(z_1, z_2, \ldots, z_k)^\perp$.*

Therefore, if the concept set consists only of a handful of non-spurious concepts, that is, semantically distinct from the confounders (which in the CLIP embedding metric translates into a small cosine similarity), we can expect the bias injected by the spurious feature to be significantly reduced.

### 4.1 Effect of Errors-in-variable

We must admit that our concept identifier can be imperfect, implying our foundational embedding consists of noisy covariates, requiring an errors-in-variables analysis [14].

Although the effect of noise on multiple covariates is complex, significant noise in any covariate influences the coefficients of all other covariates (see Appendix A.2), degrading the accuracy of the estimator.

In the case of a single noisy feature $x_K = x_K^* + u$, where $x_i$ represents the observed covariate with noise $u$, and $y = x^*\omega_* + \epsilon$ is the actual model, the least-squares estimator $\hat{\omega}$ is biased. Specifically, the estimated coefficient for this variable is attenuated by $\omega_K = \left( \frac{\sigma_K^*}{\sigma_K^* + \sigma_u^2} \right) \omega_{*K}$. Not only the estimated coefficient $\omega_K$ is increasingly shrunk by the noise towards zero, but all the other coefficients $\hat{\omega}_i$ for noiseless features are also biased proportionally to the bias in the coefficient of the noisy regressor. This interdependence means that significant noise in any covariate distorts all coefficients.

## 5 Concept sets generation and identification

While Theorem 1 refers to the properties that a concept set must satisfy, to use such a concept set, it is required that we have the concept mappings $c : \mathcal{X} \to [0,1]^{|C|}$. This can be achieved by collecting a dataset $\mathcal{X} \times \mathcal{C}$ and training a model in a supervised fashion, but this is a costly procedure. Instead, the appearance of foundational multimodal models in which one of the domains is natural language, such as CLIP [32], provides a cheap and flexible approach to concept identification [30]. We will focus on the latter approach because of its increased versatility.

From the concept set *desiderata* in Theorem 1, although powerful, direct foundational model embeddings encode information in a high-dimensional space that likely contains irrelevant information (which can act as spurious features) and noise. For example, CLIP embeddings capture syntactic and semantic nuances and may not necessarily accentuate the conceptual understandings that underpin human reasoning and decision-making processes for a given task. This hinders not only generalization but also the interpretability of the model.

### 5.1 Concept Identification

First, suppose that we already have a concept set $C$. Let us explicitly describe how to obtain the concept representation using a multimodal model.

Through the model, concepts (language) and input (e.g., images or language in the case of serialized tabular data) can be embedded into a shared space. We can then use the appropriate metric (or *similarity* measure) in this space to compute the degree of presence of a concept for each given input.

In the most common scenario, the distance consists of the cosine similarity, so if the textual concepts $C = \{t_1, \ldots, t_M\}$ and the dataset is $D = \{x_1, \ldots, x_N\}$, the concept representation $c(x)$ can be computed as

$$c_j(x) = \frac{\langle E_\mathcal{X}(x), E_T(t_j) \rangle}{||E_\mathcal{X}(x)||_2 ||E_T(t_j)||_2} \ ,$$

with $E_\mathcal{X}$ and $E_T$ being the input and text encoders respectively. This is the approach that we use in our experiment section.

## 5.2 Concept-set Generation

LLMs, especially if endowed with external sources of knowledge (such as ConceptNet [35] or Wikipedia articles), can be used to sample concepts that are commonly associated with a given task, thereby alleviating the burden of collecting domain expertise for the concept set generation, and potentially keeping the human out of the loop if necessary. See Appendix B.4 for a discussion on the choice of LLMs as base-concept samplers.

As shown in Theorem 1, the properties of the set of concepts are crucial for the effectiveness of the embedding. In particular, we should aim to balance the following objectives: (1) achieve minimal misspecification $\varepsilon$, which amounts to choosing a complete enough set of concepts to capture as much valuable information in the inputs as possible while also aiming for (2) a set of concepts as small as possible. Additionally, there is an additional practical requirement for the chosen concepts: that the model we use is good enough to identify them to minimize the error-in-variable effect.

Motivated by our theoretical *desiderata*, we propose a multistep approach.

1. Initial generation of a complete concept set that satisfies (1). To achieve that, we propose employing a language model $L$ to sample a sufficiently expressive set to achieve a sufficiently low misspecification $\varepsilon$. This can be promoted by conditioning the generation on the context task $\mathcal{C}_\mathcal{T}$ so that concepts are more relevant, which could include external knowledge such as ConceptNet and Wikipedia to promote further quality and diversity $\mathcal{C}_\mathcal{K} \subset \mathcal{C}_\mathcal{T}$. That is,

$$\tilde{c}_i \sim L(c|\mathcal{C}_\mathcal{T})$$

2. For each concept, we must ensure it can be properly identified and its presence is informative for $\mathcal{T}$. That is, $\tilde{c}_i$ should be discarded if the entropy of the concept for that task $\Omega(c_i(X))$ is below a given threshold $\Omega_m$.

3. Finally, we should remove redundant concepts[3]. The approach for this step can vary depending on the output model we use. The conceptually most straightforward approach, based on Theorem 1 consists of empirically estimating the risk $\hat{\mathcal{R}}(\hat{f}_C)$ by averaging the mean square error $\frac{1}{N}\sum_{p=1}^{N} MSE[\hat{f}_C(\cdot; \theta(\mathcal{D}_p))]$ on cross-validation samples $\{\mathcal{D}_p\}_{p=1}^N$.

All in all, the concept generation procedure can be summarized as follows. Starting from $C_0 = \emptyset$, while $\varepsilon(C_i) > \varepsilon_m$ we increase our concept set by[4]

$$C_i = C_{i-1} \cup c_i, \quad \text{where} \quad c_i \sim L(c \mid \mathcal{C}_\mathcal{T}) \quad \text{conditioned on} \quad \begin{cases} \Omega(c_i(X)) > \Omega_m \\ \hat{\mathcal{R}}(\hat{f}_{C_i}) - \hat{\mathcal{R}}(\hat{f}_{C_{i-1}}) > R_m \end{cases}$$

There are settings where we cannot reach values of $\varepsilon$ below our chosen threshold through any human concept set. For example, this can happen when there is a subset of relevant features for the task about which humans lack expertise.

In such cases, concepts can be leveraged beyond a bottleneck, which might help in any case where the misspecification term dominates. An option is to employ concepts as additional features. Another is to fit the residuals of the CBM by a black box model on the original features to enhance performance [41]. In both cases, one could use concepts that should not influence predictions by including a mutual information penalty to the models trained on the raw input.

---

[3]As per Theorem 1, a concept that does not decrease $\varepsilon$ (e.g., if it is spanned by the other concepts or orthogonal to the target function) will be detrimental to the risk since it increases the first term. In practice, when the effect on $\varepsilon$ is small, keeping such a concept should depend on the number of samples we have. In low sample regimes, we should be more aggressive in discarding it than when $n$ grows, and the term $\varepsilon$ starts to dominate.

[4]In our experiments, we keep a concept if it improves the estimated risk, i.e., $R_m = 0$, or $\hat{\mathcal{R}}(\hat{f}_{C_i}) - \hat{\mathcal{R}}(\hat{f}_{C_{i-1}}) > 0$, and we set the entropy threshold proportionally to the number of buckets $n_b$ used to estimate $\Omega$ to ensure not all samples fall into the same bucket (i.e. $\Omega_m = \frac{1}{n_b}$).

# 6 Empirical Validation

To empirically validate our insights about concept-based representations, we designed a series of experiments to validate our analysis[5]. We aim to explore the key insights from our theoretical results, including the ability to effectively leverage scarce datasets and the impact of concept-based representations on robustness to distribution shifts.

**Datasets** Our experiments utilize **CIFAR-10** [25] to test the effect of concept representations on data efficiency. This dataset is used because of its extensive use in research: its familiarity enables us to gather insights more effectively. **MetaShift** [26] is a dataset to evaluate distribution shifts in machine learning models. It comprises 12,868 subsets across 410 classes, each representing unique contexts such as "cat with sink" or "cat with fence", using data from Visual Genome [24]. The dataset's graph structure connects subsets by similarity, enabling the quantification shifts and visualization of data conflicts. Thus, we use MetaShift to evaluate model robustness across diverse shifts.

In Appendix D we discuss additional experiments, where we utilized three diverse datasets: CUB-200-2011 [37], Food-101 [7], and the Describable Textures Dataset (DTD) [9] for data efficiency analysis, CINIC-10 [11] for out-of-distribution generalization, and CIFAR-10 [25] to compare with label-free concept bottleneck models.

**Models** We use VIT-B-32 for the multimodal CLIP [32] and GPT-4 [2] as the human concept sampler. We refer to the VIT-B-32 embedding as the vanilla embedding, which acts as the natural baseline. We primarily use a linear output layer on top of either the vanilla or the concept embedding. Still, we also assess a variety of output layers to understand which concept-based inductive bias is helpful.

For all experiments, we provide a complete description, including the concept sets, in Appendix C.2.

**The effects of the concept and sample sizes.** We start by assessing the agreement between theory and practice regarding the effect of the number of training examples and the concept set size on the model's generalization to examples from the same distribution in CIFAR10, as we plot in figure 1.

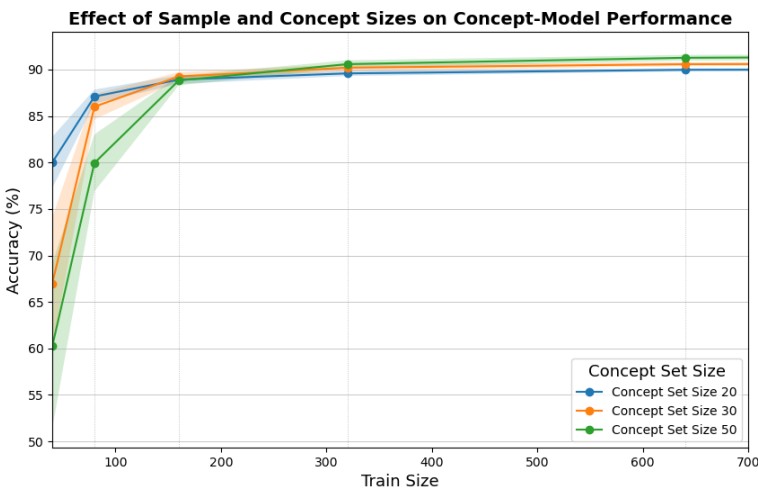

Figure 1: Scaling laws for the number of examples across different concept set sizes. As predicted by Theorem 1, smaller concept sizes $n < 100$ benefit the small sample regime, whereas larger concept sizes $n > 200$ are superior in the larger sample regime. We plot the mean and $1 - \sigma$ bars over ten seeds.

---

[5]Code can be found at https://github.com/maxruizluyten/theoretical-concepts or https://github.com/vanderschaarlab/theoretical-concepts.

As we concluded from our interpretation of Theorem 1, we expect the first term, $\left(1 - n\frac{d-|\mathcal{C}|}{(d+2)(d-1)}\right)\mathcal{R}(\hat{f}_b)$ to dominate the risk in the low data regime $n \to 0$, since in that case, the $\varepsilon$ coefficient $n\left(\frac{1}{d} + \frac{d-n}{(d+2)(d-1)}\right)$ renders the second term negligible.[6] In this domain, an increase in the sample efficiency from the first term directly depends on the concept set size $|\mathcal{C}|$, favoring smaller concept sets, which is in total agreement with the empirical results shown in figures 1 and 4.

Then, as the number of samples increases substantially $n \to \infty$, the risks start to plateau, and the misspecification, embodied by $\varepsilon$, becomes the dominant term. Thus, we expect larger concept sizes to outperform smaller ones due to a smaller $\varepsilon$. This is particularly the case given our concept set generation procedure, where smaller concept sets are subsets of the larger ones. Again, this behavior is clearly represented in figures 1 and 4.

**Robustness of Concept-embeddings to distribution shifts**  We now evaluate the concept's robustness to distribution shifts. In particular, we follow the approach of [26] in their evaluation of subpopulation shift in Section 4.2., controlling the degree of imbalance in the dataset to probe its effect on CBMs.

That is, we train and test distributions on mixtures of the same domain but where mixture weights differ between train and test. The domain comprises dog and cat images segregated by indoor and outdoor environments. In this case, pictures of outdoor cats and indoor dogs make up the minority groups, and we control the proportion of such samples while keeping $|X|$ fixed at 1700.

To align with the insights of Theorem 2, we manually verify that the concept set excludes environment-related concepts to reduce the expected confounder bias induced by the shift. Indeed, we observe that the norm of the min-max scaled projection of the concepts "indoor" and "outdoor" in our concept set $||(\langle c_i, z_1\rangle, \ldots, (\langle c_i, z_k\rangle)||_2$ has an average norm of $0.80$, small compared to the interconcept similarity, with an average of $1.65$.

Our results in figure 2 show that the concept representation yields a substantial increase in performance[7], with bigger improvements for more pronounced distribution shifts. This effect is also seen for other output models, such as k-nearest neighbors and decision trees, suggesting the concept representation effectively reduces the presence of spurious features and thus confirming that concept representations are more robust to spurious correlations.

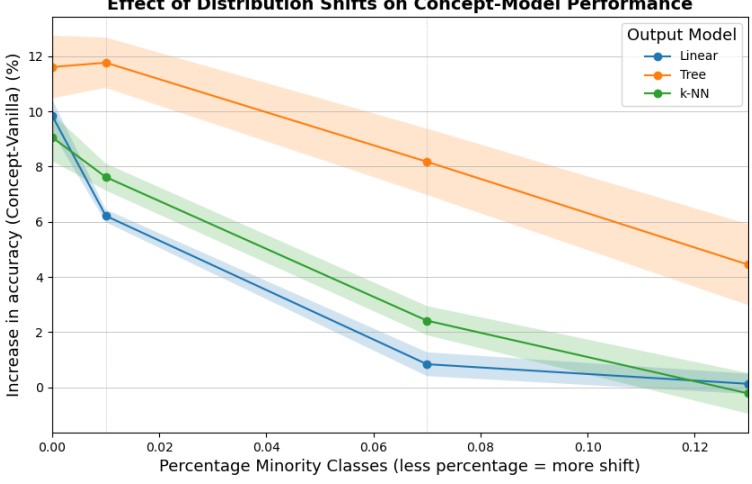

Figure 2: Concept representations reduce the effect of distribution shift over using the CLIP embedding directly, as seen by a more pronounced increase in accuracy on more imbalanced datasets. We plot the mean and $1 - \sigma$ bars over 10 seeds.

---

[6]Plus, if the concept set is appropriately chosen, the $\varepsilon$ itself should be small.
[7]Even in this sample regime, where $\varepsilon$ is the dominating term.

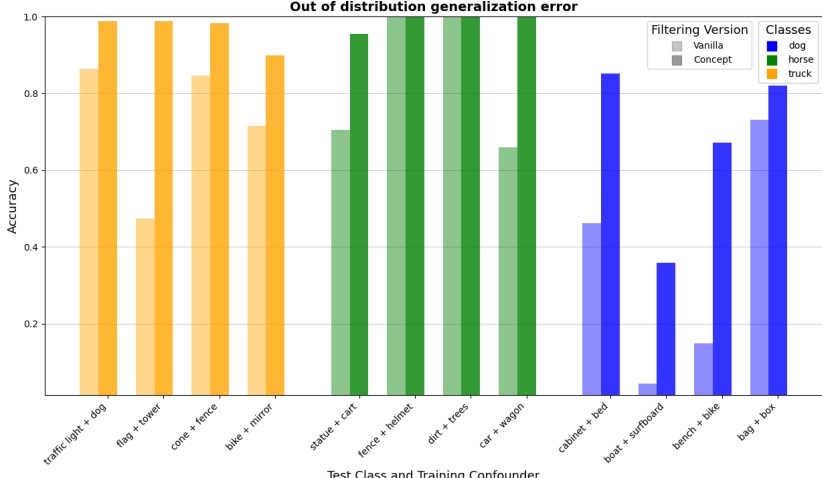

Figure 3: Concept representations consistently improve the out-of-distribution accuracy compared to directly using the CLIP embedding across different classes and distribution shifts. For three of the horse datasets, the CBM correctly identifies the presence of a horse.

**Out-of-distribution generalization of concept embeddings**   Finally, we assess the robustness of operating with the concept embedding in terms of out-of-distribution generalization. To that end, we use domain generalization datasets from Section 4.1. in MetaShift.

The datasets we use consist of three tasks: dog vs. cat, bus vs. truck, and elephant vs. horse. Each of these tasks is tested on instances from one of the two classes, in particular, on images of dogs with shelves, trucks with airplanes, and horses with barns. However, for each task, we have four different training datasets, differing by the domain in which the tested class appears. For example, in the dog vs. cat dataset, we have training sets of dogs with bags or boxes, dogs with benches or bikes, dogs with boats or surfboards, and dogs with cabinets or beds (see figure 3 for details on the other three tasks). Importantly, each of the four datasets for each class corresponds to a different degree of out-of-distribution generalization challenge, comprising a varied test bed for robustness (we refer to [26] for further details).

Figure 3 shows that for all the 12 OOD tasks, the CBM significantly outperforms the non-bottleneck baseline, typically by an accuracy gap above 10% and even surpassing a 50% increase in accuracy. We further verify the robustness of the OOD predictions of CBMs in CINIC-10 in Table 1.

## 7   Conclusion

In this work, we introduced a novel theoretical framework for understanding the properties of concept sets in Concept Bottleneck Models (CBMs). We demonstrated how these properties affect model performance, particularly in low-data regimes and under distribution shifts. Our empirical results validated these theoretical insights, showing that well-chosen concept sets improve sample efficiency and robustness in real-world scenarios. Our findings not only deepen the understanding of CBMs but also pave the way for designing more effective models. We extend the limitations and open avenues for future work discussed throughout the paper in the Appendices E and F, respectively.

## Acknowledgments and Disclosure of Funding

We want to thank the reviewers, Nicolas Huynh, Paulius Rauba, Kasia Kobalczyk, Tennison Liu, and Andrew Rashbass, for their helpful feedback. Max Ruiz Luyten is supported by AstraZeneca. This work was supported by Microsoft's Accelerate Foundation Models Academic Research initiative.

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

# A Proofs for the theoretical results

This section is divided into the lemmas we require for our proof. We start with the tools that will enable us to properly handle the spurious variables' effect in our data.

## A.1 Effects of a confounding spurious feature

We start by recovering a very recent lemma on the effect of confounding on the risk of an ordinary least squares estimator from [29]. We present it here for completeness and adapt it to our notation.

### A.1.1 Data Generating Process with a confounder

Consider a standard confounding setting in which the directed acyclic graph involves the covariates $\mathbf{X}$, the targets $\mathbf{Y}$, and the confounder covariates $\mathbf{Z}$. In this setting, $\mathbf{Z}$ is a parent of both $\mathbf{X}$ and $\mathbf{Y}$, and $\mathbf{Y}$ is a child of both $\mathbf{X}$ and $\mathbf{Z}$. In addition, we assume a Gaussian distribution and a linear setting, that is:

$$\mathbf{Y}|\mathbf{X}, \mathbf{Z} \sim \mathcal{N}_m(\beta_{y \cdot x(z)}\mathbf{X} + \beta_{y \cdot z(x)}\mathbf{Z}, \mathbf{\Sigma}_{y|x,z})$$

We use $\mathbf{\Sigma}_T$ for any random variable $T$ to denote its covariance matrix. The subscripts of the regression coefficients, as in [29], indicate the relationship between the response (before the dot) and the covariate (after the dot), considering the presence of another variable (within brackets) in the conditional mean.

### A.1.2 Effect of the confounder on the risk

Of course, the confounder is not observed, so we aim to fit the following model, where the underscore *con* stands for confounded.

$$\mathbf{Y} = \omega_{con}\mathbf{X} + \eta \quad \eta \sim \mathcal{N}_n(\mathbf{0}, \mathbf{I}_n),$$

with the least-squares estimator:

$$\hat{\omega}_{con} = \mathbf{X}^\dagger \mathbf{Y}.$$

However, in practice, due to the confounder,

$$\mathbf{Y}|\mathbf{X} \sim \mathcal{N}_n(\mathbf{A}_{y \cdot x}\mathbf{X}, \mathcal{B}^2_{y \cdot z(x)}\mathbf{\Sigma}_{z|x} + \mathbf{\Sigma}_{y|x,z}),$$

where $\mathbf{A}$ stand for the regression matrices:

$$\mathbf{A}_{y \cdot x} = \mathbf{\Sigma}_{yx}\mathbf{\Sigma}_x^{-1} = \mathcal{B}_{y \cdot x(z)}\mathbf{I}_n + \mathcal{B}_{y \cdot z(x)}\mathbf{A}_{z \cdot x}$$

**Lemma 1.** *In the above setting, the bias and variance of the least squares estimator $\hat{\omega}$ can be expressed as:*

$$Bias_Y[\hat{\omega}_{con}|\mathbf{X}] = \mathcal{B}_{y \cdot z(x)}\frac{\mathbf{X}^\top \mathbf{M}\mathbf{A}_{z \cdot x}\mathbf{X}}{\mathbf{X}^\top \mathbf{M}\mathbf{X}},$$

$$Var_Y[\hat{\omega}_{con}|\mathbf{X}] = \frac{\mathbf{X}^\top \mathbf{M}\mathbf{\Sigma}_{y|x}\mathbf{M}\mathbf{X}}{(\mathbf{X}^\top \mathbf{M}\mathbf{X})^2},$$

*where $\mathbf{M} = \mathbf{I}_n - \frac{\mathbf{1}_n\mathbf{1}_n^\top}{n}$ is the centering matrix.*

*Proof.* See Proposition 2 in [29]. $\qquad\qquad\square$

Lemma 1 implies that when both $\mathcal{B}_{y \cdot z(x)} \neq 0$ and $\mathbf{A}_{z \cdot x} \neq 0$, the estimator $\hat{\beta}_x$ is biased. Note that the expectation of the bias is proportional to both the effects of $\mathbf{Z}$ on $\mathbf{X}$ and $\mathbf{Y}$. We defer its computation to Theorem 2 in [29], but the above observation allows us to state Theorem 2.

**Theorem 3.** *In the presence of confounders $\{z_1, z_2, \ldots, z_k\} \in \bar{\mathcal{X}}^*$, given a concept $c \in \bar{\mathcal{X}}^*$, the bias in the ordinary least squares estimator satisfies*

$$||Bias(\hat{f}_C|X)||_2 \propto ||(\langle c, z_1 \rangle, \ldots, (\langle c, z_k \rangle))||_2$$

*In particular, the classifier $\hat{f}_{\{}c\}$ remains unbiased if and only if $c \in span(z_1, z_2, \ldots, z_k)^{\perp}$.*

*Proof.* We can rewrite the lemma 1 by denoting $\tilde{v} = \frac{v}{||v||_2}$ as follows:

$$||\text{Bias}_Y[\hat{\omega}_{con}|\mathbf{X}]||_2 = ||\mathbf{A}_{z \cdot x}||_2 ||\mathcal{B}_{y \cdot z(x)} \frac{\mathbf{X}^{\top} \mathbf{M} \tilde{\mathbf{A}}_{z \cdot x} \mathbf{X}}{\mathbf{X}^{\top} \mathbf{M} \mathbf{X}}||_2,$$

But $||\mathbf{A}_{z \cdot x}||_2$ corresponds, in fact, to the regression coefficient of $x(z)$. In our case, $\mathbf{A}_{z \cdot x} = (\langle c, z_i \rangle)_{i=1}^k$, which completes the proof. $\qquad\square$

## A.2 Effect of Errors-in-variable

Note that our concept identifier will not be perfect in practice, leading us to treat our foundational embedding as noisy covariates. This aligns with the errors-in-variables model [14]. Although the effect of noise across multiple covariates is complex, we can qualitatively conclude that significant noise in any covariate influences the coefficients of all other covariates, resulting in substantial degradation of the overall estimator's accuracy.

Mathematically, if $x_i = x_i^* + u_i$ represents the observed covariates with noise $u_i$, and $y = x^* \alpha^* + \epsilon$ is the true model, the least-squares estimator $\hat{\alpha}$ is biased. Specifically, for a single noisy covariate $x_K$, the attenuation effect on the estimated coefficient $\alpha_K$ is given by $\alpha_K = \left( \frac{\sigma_K^*}{\sigma_K^* + \sigma_u^2} \right) \alpha_K^*$. This indicates that the estimated coefficient $\alpha_K$ is shrunk towards zero, with the degree of attenuation increasing with the noise variance $\sigma_u^2$.

Furthermore, the bias in the estimated coefficient $\hat{\alpha}_i$ of a correctly measured regressor $x_i$ is proportional to the bias in the coefficient of the noisy regressor. This is expressed as $\hat{\alpha}_i = \alpha_i^* + \beta_i(\alpha_K^* - \alpha_K)$. This interdependence means that noise in any covariate can propagate, affecting all other coefficients ($cf.$[14]). Thus, significant noise in any covariate leads to a complex distortion of the entire regression model.

## A.3 Risk of the conceptual representation

The proof of Theorem 1 follows the same backbone as [33] and also relies on the following lemma, the proof of which we reproduce here for completeness.

**Lemma 2.** *Let $D, \Lambda \in \{0,1\}^{d \times d}$ be two diagonal matrices of rank $p$ and $q$, respectively. Let $V \in R^{d \times d}$ be an orthogonal matrix and $W \in R^{d \times d}$ a Haar distributed random matrix. If $P = WDW^T, Q = V\Lambda V^T$, then*

$$\mathbb{E}_W[PQP] = \frac{p}{d(d-1)(d+2)} [q(d-p)I_{d \times d} + [d(p+1) - 2]Q].$$

*Proof.* Since $V$ is an arbitrary orthogonal matrix and $W$ is Haar distributed, without loss of generality, we can assume $\Lambda = \text{diag}(\mathbf{1}_q, \mathbf{0}_{d-q})$, and $D = \text{diag}(\mathbf{1}_p, \mathbf{0}_{d-p})$. By definition of the Haar distribution, we can denote $U = V^T W$, also Haar distributed. Therefore,

$$\mathbb{E}_W[PQP] = \mathbb{E}_{W \sim Haar}[WDW^T V \Lambda V^T W D W^T]$$
$$= V \mathbb{E}_{U \sim Haar}[UDU^T \Lambda U D U^T] V^T$$
$$= V \mathbb{E}_{U \sim Haar}[A^T A] V^T,$$

where we have defined $A = \Lambda U D U^T$. Looking closely at the entries of $A$ a quick computation yields, denoting $\tilde{u}_i = (u_{i1}, \ldots, u_{ip})$:

$$A = \begin{pmatrix} \langle \tilde{u}_1, \tilde{u}_1 \rangle & \cdots & \langle \tilde{u}_1, \tilde{u}_d \rangle \\ \vdots & \ddots & \vdots \\ \langle \tilde{u}_q, \tilde{u}_1 \rangle & \cdots & \langle \tilde{u}_q, \tilde{u}_d \rangle \\ 0 & \cdots & 0 \\ \vdots & \ddots & \vdots \\ 0 & \cdots & 0 \end{pmatrix}.$$

Thus, $(A^T A)_{i,r} = \sum_{k=1}^{q} \langle \tilde{u}_i, \tilde{u}_k \rangle \langle \tilde{u}_r, \tilde{u}_k \rangle$, and so, $\mathbb{E}_U[A^T A]$ can be readily computed if we have the fourth moments of the entries in $U$. Indeed,

$$\mathbb{E}_U \left[ (A^T A)_{i,r} \right] = \sum_{\alpha=1}^{q} \sum_{j,s=1}^{p} \mathbb{E}_U \left[ u_{ij} u_{rs} u_{\alpha j} u_{\alpha s} \right].$$

Lemma 9 from [8] provides such 4th moments: if $i \neq r$, then $\mathbb{E}_U[(A^T A)_{i,r} = 0$, and if $i = r$, then

$$\mathbb{E}_U \left[ u_{ij} u_{is} u_{\alpha j} u_{\alpha s} \right] = \frac{1}{d(d-1)(d+2)} \left[ d\delta_{js} + d\delta_{i\alpha} + (d-2)\delta_{i\alpha}\delta_{js} - 1 \right].$$

Consequently, the expectation yields a diagonal matrix with two blocks of eigenvalues:

$$\mathbb{E}_U \left[ (A^T A)_{i,i} \right] = \frac{p}{d(d-1)(d+2)} \left[ (d-p) I_{d\times d} + [d(p+1) - 2] \Lambda \right].$$

Since $\mathbb{E}_W[PQP] = V \mathbb{E}_U \left[ (A^T A) \right] V^T$, $VV^T = I_{d \times d}$, and $V \Lambda V^T = Q$, the result follows. $\qquad \square$

Next, we follow the steps of [33] to arrive to Theorem 1

**Theorem 4.** *Let $\bar{\mathcal{X}} = \mathbb{R}^d, \mathcal{C} = \mathbb{R}^{|\mathcal{C}|}, \mathcal{Y} = \mathbb{R}^l$ And assuming that $\mathbb{P}$ is an isotropic distribution, where $X = \{x_i\}_{i=1}^{n} \sim_{iid} \mathbb{P}$ is full rank, and our labels are linear on the covariates, i.e., $y_i = \omega_* x_i$ then the risk of the concept-based model $\mathcal{R}(\hat{f}_C)$ is given by*

$$\mathcal{R}(\hat{f}_C) = \left( 1 - n \frac{d - |\mathcal{C}|}{(d+2)(d-1)} \right) \mathcal{R}(\hat{f}_b) + n \left( \frac{1}{d} + \frac{d-n}{(d+2)(d-1)} \right) \varepsilon,$$

*where $\varepsilon = ||\omega_* C^\perp|||_F^2$, and $C^\perp$ stands the orthogonal projection operator onto the orthogonal complement of the concept vector's span.*

*Proof.* Given a matrix $M \in \mathbb{R}^{k \times p}$ full rank $p$, we denote $M^{||} := M^\dagger M$ the projection onto its column space, and $M^\perp = I_{d \times d} - X^{||}$ the projection into the corresponding orthogonal space.

Then, denoting $\hat{\omega}_C$ the ordinary mean square estimator of the weights we have:

$$\hat{\omega}_C C = Y(CX)^\dagger C = \omega_* XX^\dagger C^\dagger C = \omega_* X^{||} C^{||}.$$

Therefore,

$$\begin{aligned} \hat{\omega}_C C - \omega_* &= \omega_* \left( X^{||} C^{||} - I_{d \times d} \right) \\ &= \omega_* \left[ \left( X^{||} - I_{d \times d} \right) C^{||} + C^{||} - I_{d \times d} \right] \\ &= -\omega_* \left[ X^\perp C^{||} + C^\perp \right]. \end{aligned}$$

Using the cyclic property of the trace, the risk is given by

$$\begin{aligned} \mathcal{R}(\hat{\omega}_C C) &= \mathbb{E}_{X,x} \left[ ||\hat{\omega}_C C x - \omega_* x||^2 \right] \\ &= \text{Tr} \left( \mathbb{E}_{X,x} \left[ \omega_* \left( X^\perp C^{||} + C^\perp \right) x x^T \left( X^\perp C^{||} + C^\perp \right)^T \omega_*^T \right] \right) \\ &= \text{Tr} \left( \mathbb{E}_X \left[ \omega_* \left( X^\perp C^{||} + C^\perp \right) \left( X^\perp C^{||} + C^\perp \right)^T \omega_*^T \right] \right). \end{aligned}$$

Since projections are idempotent and $C^\perp C^{||} = C^{||} C^\perp = \mathbf{0}$,

$$\left(X^\perp C^{||} + C^\perp\right)\left(X^\perp C^{||} + C^\perp\right)^T = X^\perp C^{||} C^{||} X^\perp + X^\perp C^{||} C^\perp + C^\perp C^{||} X^\perp + C^\perp C^\perp$$
$$= X^\perp C^{||} X^\perp + C^\perp,$$

and as a consequence, that

$$\mathcal{R}(\hat{\omega}_C C) = \text{Tr}\left(\mathbb{E}_X\left[\omega_*\left(X^\perp C^{||} X^\perp + C^\perp\right)\omega_*^T\right]\right)$$
$$= \text{Tr}\left(\omega_*\left(\mathbb{E}_X\left[X^\perp C^{||} X^\perp\right] + C^\perp\right)\omega_*^T\right).$$

$X^\perp, C^{||}$ are both projections and since $X$ follows an isotropic distribution according to our assumptions, its right singular vectors, which coincide with the eigenvectors of $X^\perp$ are Haar distributed. Therefore, we can apply the lemma 2 with $p = d - n, q = |\mathcal{C}|$ we obtain that

$$\mathbb{E}_X\left[X^\perp C^{||} X^\perp\right] = \left(1 - \frac{n}{d}\right)\left[\frac{|\mathcal{C}|n}{(d-1)(d+2)}I_{d\times d} + \left(1 - \frac{dn}{(d-1)(d+2)}\right)C^{||}\right].$$

Recalling $C^{||} = I_{d\times d} - C^\perp$ and rearranging the terms we obtain

$$\mathbb{E}_X\left[X^\perp C^{||} X^\perp\right] + C^\perp = \left(1 - \frac{n}{d}\right)\left(1 - \frac{n(d - |\mathcal{C}|)}{(d+2)(d-1)}\right)I_{d\times d}$$
$$+ \left(\frac{n}{d} + \frac{n(d-n)}{(d+2)(d-1)}\right)C^\perp.$$

Lastly, we use $\mathcal{R}(\hat{\omega}_t) = \left(1 - \frac{n}{d}\right)$ (see [5]) and our definition of $\varepsilon := \text{Tr}\left(\omega_* C^\perp \omega_*^T\right)$ to conclude that

$$\mathcal{R}(\hat{\omega}_C C) = \left(1 - n\frac{d - |\mathcal{C}|}{(d+2)(d-1)}\right)\mathcal{R}(\hat{\omega}_t) + n\left(\frac{1}{d} + \frac{d - n}{(d+2)(d-1)}\right)\varepsilon,$$

$\square$

# B  Extended Discussions

In this section, we expand on specific details of the paper that, albeit non-essential for the main text, might be of interest to specific readers.

## B.1  Further details on the Concept Set definition

To ensure that the composition is well defined, we need to impose one of the following two conditions: either for the output probe or for the concept set. Either (1) if concept sets are unordered, then $\mathcal{H}_\mathcal{C}$ must be a complete set of maps under permutation symmetry, that is, for any map $g \in \mathcal{H}_\mathcal{C}$ and any permutation $\sigma$ of $\mathcal{C}$, $g \circ \sigma \in \mathcal{H}_\mathcal{C}$, and the learning method is also invariant under feature permutations, which holds for the output methods considered in the paper; alternatively, (2) $\mathcal{C}$ is a concept sequence, not a set. In the first case, the composition is well defined since it is equivalent across all orderings of the elements; in the second, $\cup$ should be substituted by $\times$.

We note that the natural models used as probes—such as linear regression, kernel regression, fully connected neural networks, decision trees, and k-nearest neighbors, among others—constitute parameterized hypothesis spaces that form a complete set of maps under permutation symmetry. Additionally, the learning methods employed are invariant under feature permutations, provided that isotropic initialization is used and mild assumptions regarding the optimization algorithm and loss function are met. Consequently, these models and their associated learning procedures maintain structural integrity and invariance when input features are permuted, ensuring that the composition remains well-defined across different concept orderings.

## B.2  Non-Concept characteristic functions

To elucidate the relevance of the properties in Definition 1 that distinguish *Concept sets* from other sets of characteristic functions, it is helpful to consider non-concept characteristic functions. We begin with a specific example before advancing to a more formal characterization.

Humans tend to develop concepts that are relevant to their areas of interest, which are indicative of their underlying task $\mathcal{T}$. Despite being exposed to a wide range of occurrences, individuals do not cultivate concepts outside their domain of interest. For instance, consider filmography-related concepts such as a "Jump Cut," where two sequential shots of the same subject are captured from slightly varying camera positions, or a "Dutch Tilt," involving the intentional tilting of the camera to create a sense of disorientation. According to Definition 1, people not interested in filmography would not incorporate these characteristic functions into their concept set. This omission occurs because such characteristics add negligible expressiveness to their existing concept set, meaning that the parameter $N_0$ approaches infinity without significantly enhancing model-aware inductive bias. Consequently, $N_1$ becomes less than $N_0$, which does not satisfy the definition.

In more abstract terms, a characteristic function $c$ that does not contribute additional valuable information beyond the existing concept set $\mathcal{C}$ for the task $\mathcal{T}$ exhibits low expressiveness, indicated by a large $N_0$, without a corresponding increase in $N_1$. This scenario arises when the target variables are conditionally independent of $c$ given $\mathcal{C}$. Examples include concepts that are irrelevant to the task at hand (e.g., the presence of tinted glass when predicting attributes of a Model X vehicle) or concepts that are entirely redundant with those already present in $\mathcal{C}$.

Moreover, low expressiveness can result when the output model is incapable of capturing the correlation introduced by $c$. For example, a characteristic function $c$ could aid in clustering the input distribution into two concentric spheres, but a linear model would fail to take advantage of this structure effectively. Furthermore, even if $c$ provides information that is theoretically representable, the inductive bias of the learning method may disfavor the optimal hypothesis corresponding to this representation. In such cases, despite the presence of informative characteristic functions, $N_0$ remains large, indicating low expressiveness.

## B.3  Concepts as a stable communication interface

We expand on the flexibility of Concept-Based Models (CBMs), highlighting properties that can facilitate future advancements.

Consider the objective of learning a target function $f^* : \mathcal{X} \rightarrow \mathcal{Y}$. In a CBM framework, this involves a concept representation $\mathfrak{C}^* : \mathcal{X} \rightarrow \mathcal{R}_\mathcal{C}$ followed by a mapping $f^*_\mathcal{C} : \mathcal{R}_\mathcal{C} \rightarrow \mathcal{Y}$ from concepts to targets. By maintaining a fixed concept set $\mathcal{C}$ and its associated representation $\mathcal{R}_\mathcal{C}$, we allow the replacement of the concept mapping estimator $\hat{\mathfrak{C}}$ with an alternative $\tilde{\mathfrak{C}}$ without altering the probe $\hat{f}_\mathcal{C}$. This substitution is viable as long as the concept representation $\mathcal{R}_\mathcal{C}$ remains consistent in dimensionality and semantic meaning. For example, upgrading from a CLIP-based concept representation to a more advanced CLIP2 preserves the probe's applicability.

However, probes are not interchangeable between fundamentally different concept representations (e.g., raw CLIP vs. CLIP2) due to potential discrepancies in metrics or dimensionality. This emphasizes the role of $\mathcal{R}_\mathcal{C}$ as a stable interface with fixed characteristics.

Furthermore, CBMs facilitate the harmonization of datasets with varying features or modalities $\mathcal{X}_1, \ldots, \mathcal{X}_k$ (such as images from different sensors). By projecting each input $\mathcal{X}_i$ into the common concept space $\mathcal{R}_\mathcal{C}$ through mappings $\mathfrak{C}^*_i : \mathcal{X}_i \rightarrow \mathcal{R}_\mathcal{C}$, CBMs ensure semantic consistency across diverse datasets. This unified representation simplifies the integration and analysis of multi-modal data, leveraging the fixed semantic meaning of the concept set.

## B.4 Use of Large Language Models for Concept Generation

The primary objective of our method and experiments is to illustrate and validate our theoretical insights, which remain agnostic to the concept sampling method. Although large language models (LLMs) are utilized for concept generation in our experiments, this choice does not compromise the validity or generality of our theoretical contributions. Our theoretical insights are independent of the concept generation method, which emphasizes the versatility and applicability of our insights regardless of concepts being derived from LLMs, human experts, structured knowledge bases, or a combination thereof.

### B.4.1 Alternative Concept Sampling

We employ LLMs to streamline the process by reducing the need for extensive human expertise and to provide task-specific concepts flexibly. LLMs have been shown to be valuable samplers/generators, especially as part of a more complex system designed to be robust against potential lower-quality samples (e.g., [4, 17, 34]), but alternative methods are equally viable. These alternatives include:

- **Human-Generated Concepts**: Concepts manually curated, tailored to specific tasks.
- **Concept Graphs**: Using structured knowledge bases such as ConceptNet [35] to derive relevant concepts.
- **Correlated Text Corpora**: Leveraging patterns and correlations within large text datasets like Wikipedia to identify meaningful concepts.
- **Hybrid Approaches**: Enhancing LLM-generated concepts by conditioning their context on external knowledge sources, such as *Concept Graphs* or *Text Corpora*, thus improving their relevance and quality, as mentioned in the main text.

### B.4.2 Quality Control through Rejection Sampling

We also note that our method can be regarded as a *rejection-sampling loop* that ensures the quality of the concepts generated and mitigates the potential limitations associated with the reliance on LLMs for concept sampling. It does so by rejecting concepts that do not meet predefined theoretical standards and selecting only those concepts that significantly contribute to the model-aware inductive bias, thereby maintaining the integrity of the concept set.

This approach thus fosters the inclusion of only relevant and high-quality concepts, regardless of the base generation method.

# C Experiment details

## C.1 Assets used

**Datasets** For CIFAR10 we use CIFAR-10 python version [25], and for MetaShift (MIT License) we follow the scripts provided by the MetaShift Repo [26].

**Models** For generating concept sets, we use GPT 4-turbo-2024-04-09 [2] through the ChatGPT web app. For jointly embedding images and text, we use CLIP ViT-B/32 [32].

## C.2 Compute Resources Used for the Experiments

The computational resources used are consistent across all experiments and are detailed below to facilitate reproducibility. Specifically, an Azure Machine NC96ads A100 v4 was used with the following specifications:

**GPU Details:**

- **Type of Compute Workers:** A single NVIDIA V100 GPU was used.
- **Memory:** Each NVIDIA V100 GPU is equipped with 80 GiB of memory.
- **Time of Execution:** Each experiment runs in less than 5 hours on a single GPU.

The total computing time between experiments was approximately 20 hours. No further computing was required for the project, except for dummy runs intended to verify the code's correct functioning on an M2 Pro MacBook Pro with 32 GiB of memory.

**Additional System Information:**

- **CPU:** AMD EPYC 7V13 64-Core Processor with 96 CPUs at 3.09 GHz.
- **RAM:** 866 GiB total, 23 GiB used, 832 GiB free, 61 GiB swap memory.

## C.3 Concept set generation before filtering

For the generation of concept sets, instead of keeping or discarding a concept before generating a new one, we first generated a pool of concepts, which we then sampled sequentially and filtered the subset of interest. To generate the initial pool, we started by providing the following prompt to GPT 4-turbo-2024-04-09:

```
I am building a classifier between {Class_Names} on the {Dataset_Name}
dataset.  I want to use a concept-bottleneck model to predict these classes.

To that end, I want you to propose a set of {N} concepts for each
class that can help me differentiate between {Class_Names} for the
concept-bottleneck model.

Propose only concepts that can be identified from an image without sound
or video.
```

From the result of this prompt, we trimmed the concepts (e.g., "animal with floppy ears" → "floppy ears") to make them more suitable for CLIP embedding. The resulting initial pool of concepts for each experiment is provided below in the detailed description.

## C.4 Detailed description: Effects of concept and sample sizes

**Dataset information**

The dataset used in this experiment is CIFAR-10, which consists of 10 classes: Airplane, Automobile, Bird, Cat, Deer, Dog, Frog, Horse, Ship, and Truck.

The train sizes evaluated are [40, 80, 160, 320, 640, 1280, 2560, 5120, 10000] samples, and the test size is fixed at the full 10,000 test samples.

**Configuration**

The embedding model used is CLIP ViT-B/32. The preprocessing pipeline utilizes torchvision.transforms in several steps. It begins by resizing the image to 224x224 pixels with bicubic interpolation, ensuring smooth resizing. It then performs a center crop to retain the central 224x224 pixels, focusing on the most significant part of the image. The image is then converted to a tensor, scaling the pixel values from [0, 255] to [0.0, 1.0]. Finally, the tensor is normalized using specific mean and standard deviation values for each RGB channel, standardizing the input data.

The output models evaluated are based on linear regression on the one-hot encoded labels, with coefficients determined using an exact solver.

The random seeds used for this evaluation are [42, 43, 44, 45, 46, 47, 48, 49, 50, 51].

**Unfiltered Concept Set**

The initial pool of concepts generated by GPT-4 from which we sample as described in Section 5 is:

- **Airplane:** Fixed wings, Jet engines, Propellers, Cockpit windows, Landing gear, Tail fin, Airplane fuselage, Winglets, Long, narrow wings, Round windows, Brightly colored livery, Nose cone, Smooth metallic body, Horizontal stabilizers, Underwing pods, Delta wings, Commercial airline logos, Contrails, Navigation lights, Turbofan engines.

- **Automobile:** Four wheels, Headlights, Tail lights, Rearview mirrors, Grille, Doors, Windows, Bumpers, Roof rack, Alloy wheels, License plate, Windshield wipers, Exhaust pipe, Hood, Trunk, Side mirrors, Car brand logos, Spoiler, Roof antenna, Fog lights.

- **Bird:** Feathers, Beak, Wings, Tail feathers, Claws, Perching behavior, Colorful plumage, Small size, Sharp eyesight, Nesting, Beady eyes, Flight patterns, Hollow bones, Singing or chirping, Flapping wings, Broad wingspan, Crest or crown, Scaly legs, Bird nest, Smooth beak.

- **Cat:** Sharp, pointy ears, Rounded eyes, Fine, soft fur, Short, pointed snout, Fine whiskers, Slender and agile body, Compact paws, Grooming behavior, Arching back stretch, Retractable claws, Slit pupils, Purring, Curved claws, Lithe movement, Long tail, Flexible spine, Night vision, Whisker pads, Silent movement, Playful behavior.

- **Deer:** Antlers, Slender legs, Hooves, Brown fur, White tail, Large ears, Doe eyes, Grazing behavior, Spotted coat, Small tail, Rounded snout, White underbelly, Graceful movement, Herd behavior, Muzzle, Alert posture, Small antlered males, Thin, long legs, Winter coat, Sleek body.

- **Dog:** Floppy ears, Bushy tail, Curly tail, Almond-shaped eyes, Smooth short fur, Curly fur, Wiry fur, Long, narrow snout, Thick whiskers, Large body, Large paws, Wagging tail, Barking, Droopy ears, Loyal posture, Collar, Panting, Playful behavior, Sniffing, Guarding behavior.

- **Frog:** Smooth, moist skin, Webbed feet, Bulging eyes, Long hind legs, Jumping, Sitting posture, Wide mouth, Green color, Camouflaged skin, Sticky tongue, Croaking, Slender fingers, Tadpole stage, Water habitat, Streamlined body, Short front legs, Bug catching, Wide head, Swimming ability, Amphibious nature.

- **Horse:** Mane, Tail, Hooves, Long legs, Galloping, Large eyes, Long snout, Strong build, Grazing behavior, Smooth coat, Upright ears, Bridle, Saddle, Neighing, Herd behavior, Muscular body, White blaze, Erect posture, Speed, Stirrup.

- **Ship:** Hull, Deck, Sails, Masts, Bow, Stern, Portholes, Lifeboats, Anchor, Smokestacks, Rigging, Cabins, Cargo containers, Radar, Funnels, Bridge, Naval flag, Propeller, Sailors, Lifebuoys.

- **Truck:** Large wheels, Cargo bed, Cab, Grille, Headlights, Side mirrors, Exhaust pipe, Large size, Rearview mirrors, Mud flaps, Trailer hitch, Step-up rails, Heavy-duty tires, Flatbed, Bull bars, Utility lights, Roof lights, Storage compartments, Ladder racks, Commercial decals.

**Filtered Concept Sets**

Three concept set sizes are evaluated: 20 concepts, 30 concepts, and 50 concepts.

The filtering method described in Section 5 is used, which iteratively draws one concept for each class, the one that minimizes the empirical risk in five seeds over a 50/50 CIFAR-10 training set split. The entropy is estimated through 100 bins, and a lower threshold of $0.02$ ensures that not all activations are essentially equivalent.

### C.5 Detailed description: Robustness to distribution shifts

**Dataset Information**

The dataset used in this experiment is MetaShift, focusing on the "Cat vs. Dog" setting, as described in [26]. The percentages of minority classes evaluated are [0, 0.01, 0.07, 0.13]. The random seeds used for this evaluation are [42, 43, 44, 45, 46, 47, 48, 49, 50, 51].

**Configuration**

The embedding model used is CLIP ViT-B/32. The preprocessing pipeline is the same as in the above experiment.

The evaluated output models include a linear regressor, a decision tree, and nearest neighbors, all with the standard hyperparameters from `sklearn` and trained on one hot encoded label.

**Unfiltered Concept Set**

The initial set of concepts generated by GPT-4 from which we sample is: Sharp, pointy ears; Floppy ears; Rounded ears; Pointy ears; Thin tail with a tapering end; Bushy tail; Curly tail; Almond-shaped eyes; Round eyes; Fine, soft fur; Smooth short fur; Curly fur; Wiry fur; Short, pointed snout; Long, narrow snout; Flat snout; Fine whiskers; Thick whiskers; Slender and agile body; Large body; Compact paws; Large paws; Grooming behavior; Arching back stretch; Wagging tail.

**Filtered Concept Sets**

The filtering method described in Section 5 is used, which iteratively draws a concept and keeps it if it decreases the empirical risk in five seeds (threshold of 0). We used a 70/30 split for empirical risk estimation. The entropy is estimated through 10 bins, and a lower threshold of $0.1$ ensures that not all activations fall into the same bucket.

### C.6 Detailed description: Out-of-distribution generalization

**Dataset information**

The datasets used in this experiment are the MetaShift domain generalization datasets; see [26]. That is,

**Cat vs. Dog**  The "Cat vs. Dog" dataset has the following four different training configurations, all evaluated on the same test set that contains images of dogs on a shelf.

- **Configuration 1**: Cat on a sofa or bed, and Dog in a cabinet or bed.
- **Configuration 2**: Cat on a sofa or bed, and Dog in a bag or box.
- **Configuration 3**: Cat on a sofa or bed, and Dog on a bench or with a bike.
- **Configuration 4**: Cat on a sofa or bed, and Dog on a boat or with a surfboard.

**Bus vs. Truck**  The "Bus vs. Truck" dataset has the following four different training configurations, all evaluated on the same test set containing images of trucks with airplanes.

- **Configuration 1**: Bus with clock or traffic light, and Truck with cone or fence.
- **Configuration 2**: Bus with clock or traffic light, and Truck with bike or mirror.

- **Configuration 3**: Bus with clock or traffic light, and Truck with flag or tower.
- **Configuration 4**: Bus with clock or traffic light, and Truck with traffic light or dog.

**Elephant vs. Horse**   The "Elephant vs. Horse" dataset has the following four different training configurations, all evaluated on the same test set containing images of horses in a barn.

- **Configuration 1**: Elephant with fence or rock, and Horse in dirt or with trees.
- **Configuration 2**: Elephant with fence or rock, and Horse with fence or helmet.
- **Configuration 3**: Elephant with fence or rock, and Horse with car or wagon.
- **Configuration 4**: Elephant with fence or rock, and Horse with statue or cart.

### Configuration

The embedding model used is CLIP ViT-B/32. The preprocessing pipeline is the same as in the two previously described experiments.

The output model consists of linear regression on the one-hot encoded labels, with coefficients determined using an exact solver.

### Unfiltered Concept Set

The initial pool of concepts generated by GPT-4 from which we sample as described in Section 5 is:

- **Cat vs. Dog**: sharp, pointy ears; floppy ears; rounded ears; pointy ears; thin tail with a tapering end; bushy tail; curly tail; almond-shaped eyes; round eyes; fine, soft fur; smooth short fur; curly fur; wiry fur; short, pointed snout; long, narrow snout; flat snout; fine whiskers; thick whiskers; slender and agile body; large body; compact paws; large paws; grooming behavior; arching back stretch; wagging tail.
- **Bus vs. Truck**: Long and tall; Multiple axles; Trailer; Large, continuous rows of windows; Passenger seating; Cargo space; Grille and large headlights; Cargo doors; Passenger doors and emergency exits; Commercial text and logos; Public transit text and logos; Uniform vehicle color scheme; Varied vehicle color scheme; Air conditioning units; Visible cargo.
- **Elephant vs. Horse**: large overall size; thick skin texture; large and fan-shaped ears; tusks; trunk; wispy tail with sparse hair; bulky and robust body build; mane; small eyes relative to body size, placed more to the side of the head; large, round feet with visible toenails; short and thick neck; thick legs; uniform color without markings.

### Filtered Concept Sets

We used the same approach as in the distribution shift experiments described above.

# D Additional Experiments

In this appendix, we discuss additional experiments to illustrate our theoretical insights further by evaluating concept bottleneck models (CBMs) on additional datasets and baselines. Specifically, we discuss:

1. **Data-Efficiency Experiments** on three datasets: CUB-200-2011 [37], Food-101 [7], and the Describable Textures Dataset (DTD) [9], to evaluate the performance of CBMs across varying sample sizes.

2. **Out-of-Distribution Generalization** on the CINIC-10 dataset [11], to assess the robustness of CBMs to distribution shifts.

3. **Comparison with Label-Free Concept Bottleneck Models** [30] on CIFAR-10, benchmarking our method against a relevant baseline from the literature.

## D.1 Data-Efficiency Experiments

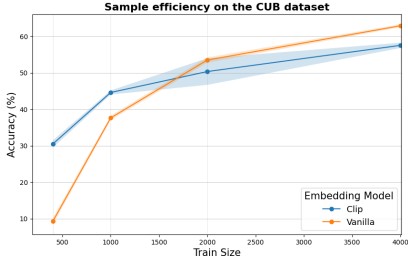

(a) Performance comparison of a linear probe on the CLIP representation vs. our CBM across different training sizes on the CUB-200-2011 dataset.

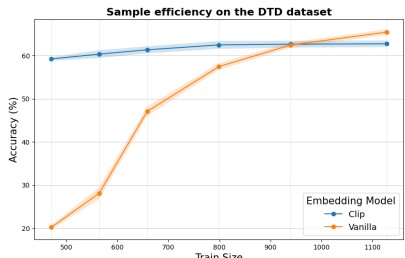

(b) Performance comparison of a linear probe on the CLIP representation vs. our CBM across different training sizes on the Food-101 dataset.

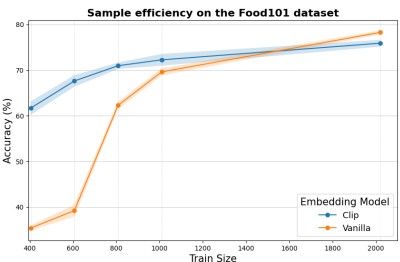

(c) Performance comparison of a linear probe on the CLIP representation vs. our CBM across different training sizes on the DTD dataset.

Figure 4: Comparison of a linear probe on the CLIP [32] representation against our CBM across different training sizes for three datasets: CUB-200-2011 [37] in Figure 4a, Food-101 [7] in Figure 4c, and DTD [9] in Figure 4b. This figure illustrates the performance of both models across varying data availability, highlighting the increased data efficiency of CBMs in the low sample regime and the effect of misspecification in the large sample regime (see Theorem 1).

### D.1.1 Experimental Setup

To evaluate the data efficiency of our CBM approach, we conducted experiments on three diverse datasets.

- **CUB-200-2011** [37]: A fine-grained dataset containing 200 classes of bird species, with 11,788 images annotated with 312 binary attributes (concepts).
- **Food-101** [7]: A dataset with 101 food categories comprising 101,000 images.

- **Describable Textures Dataset (DTD)** [9]: A texture database with 47 classes and 5,640 images, annotated with attributes describing the textures.

For each dataset, we compared the performance of:

- **Baseline**: A linear classifier trained directly on the CLIP [32] image embeddings.

- **Our CBM**: A concept bottleneck model where we first predict concepts from the image embeddings and then use these predicted concepts to classify the images.

We varied the number of training samples to assess performance in different sample size regimes. For each training size, we randomly sampled the corresponding number of examples (with balanced classes) and trained the models five times with different random seeds to obtain averaged results.

### D.1.2   Results and Discussion

The results are summarized in Figure 4. We observe the following trends:

- **Small Sample Regime**: In the low-data regime ($< 10$ samples per class), our CBM outperforms the baseline in all three datasets. This supports our theoretical insight from Theorem 1, which predicts that in the small sample regime, the first term of the risk dominates, and the structured approach of the CBM leads to increased sample efficiency.

- **Large Sample Regime**: In the larger-data regime, the baseline begins to outperform the CBM. This reversal is explained by the increasing influence of the approximation error term, encapsulated in the $\varepsilon$ coefficient in our theoretical analysis. As the sample size grows, the information loss due to the bottleneck becomes more significant, determining the asymptotic performance gap between the CBM and the baseline.

These results reaffirm the insights of Theorem 1, demonstrating how the trade-off between sample efficiency and misspecification manifests itself in practice across different data sets. The CBM's advantage in low-data settings highlights its potential for applications where labeled data are very scarce (e.g., rare diseases).

## D.2   Out-of-Distribution Generalization on CINIC-10

### D.2.1   Experimental Setup

To evaluate the robustness of CBMs to distribution shifts, we conducted an out-of-distribution (OOD) generalization experiment using the CINIC-10 dataset [11]. CINIC-10 is derived from both CIFAR-10 [25] and ImageNet [18], providing a testbed for OOD evaluation.

- **Training Data**: We trained models on a subset of the CIFAR-10 training set, with 750 images across 10 classes.

- **Test Data**: We evaluated the models in ImageNet images that belong to subclasses of the original CIFAR-10 classes to introduce a distribution shift between the training and test data.

We compared the performance of the baseline linear classifier trained on CLIP image embeddings against our CBM, using predicted concepts for classification.

### D.2.2   Results and Discussion

The results are presented in Table 1. Our CBM demonstrates superior OOD generalization performance compared to the baseline. This improved OOD performance aligns with our previous results on the 12 datasets from the MetaShift collection, as discussed in Figure 3. The CBM's reliance on intermediate concepts, which can be more stable under distribution shifts and potentially remove spurious features and confounders, as suggested by Theorem 2, contributes to its robustness.

| Model | Accuracy (%) |
|---|---|
| Vanilla | $70.4 \pm 1.2$ |
| **CBM** | $\mathbf{78.8 \pm 0.7}$ |

Table 1: Out-of-distribution generalization comparison on CINIC-10 [11] between a linear probe on the CLIP [32] embedding and our CBM method.

### D.3 Comparison with Label-Free Concept Bottleneck Models

#### D.3.1 Experimental Setup

We compared our curation of concept sets with the Label-Free Concept Bottleneck Model [30] (LF-CBM) method on the CIFAR-10 dataset.

For a fair comparison of the effect of the concept set selection, we made the following modifications.

- **Concept Set Selection**: We used the same sampling mechanism for concept selection, sampling concepts $c_i \sim L(c \mid C_T)$ using a fixed language model $L$ and context $C_T$ (i.e., prompt).
- **Image Embeddings**: Both methods used the `CLIP ViT-B/32` model to obtain image embeddings.
- **Classifier Architecture**: A linear classifier (probe) was trained on top of the concept embeddings.

We varied the training sample sizes and averaged the results over five random seeds.

#### D.3.2 Results and Discussion

Figure 5 illustrates the performance comparison between these two approaches, which highlights the critical role of concept set curation through the consistent performance gap of more than $7.3\%$ across all training set sizes and up to $17.7\%$.

These results highlight the relevance of considering the properties of concept sets when using CBMs, reinforcing the value of our theoretical contributions.

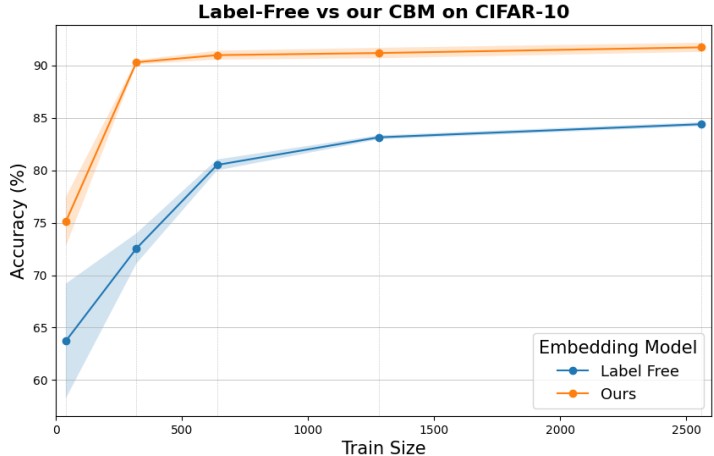

Figure 5: Performance comparison between our concept set curation method and the Label-Free Concept Bottleneck Model (LF-CBM) [30] on CIFAR-10 [25]. The results demonstrate a consistent performance gap, with our method outperforming LF-CBM by more than $7.3\%$ across all training set sizes and reaching up to $17.7\%$. This comparison underscores the impact of effective concept set selection in concept bottleneck models, validating the importance of our theoretical insights.

# E  Extended Limitations

## E.1  Notable assumptions

**The CBM Assumption**   We make the implicit assumption that human concepts have predictive power for ML models, as discussed in Section 3.

**Concepts as a linear space**   Although the assumption that concepts correspond to directions in a high-dimensional space holds empirically (e.g., for the CLIP embedding [21, 41, 6]), it may break when inspected closely. For example, this assumption cannot capture hierarchical relationships between concepts. Investigating alternative representations and methods to identify and utilize hierarchical concepts would enhance the robustness and interpretability of CBMs.

**Input distribution for Theorem 1**   We assume that the inputs are distributed isotropically for analytical tractability. Although it might not hold in some real-world distributions, this assumption includes all distributions whose density depends only on the distance to the origin, thereby being quite flexible. Additionally, we observe in section 6 that the qualitative behavior in real-world tasks aligns with the results from the theorem derived under these assumptions.

## E.2  Other limitations

**Interaction of effects**   The proposed theory addresses the analysis of in-domain generalization, the impact of confounding, and error-in-variables separately. Still, one could explore considering all of them simultaneously and their interactions. Integrating these aspects into a unified theoretical framework remains an open challenge and could provide a more comprehensive understanding of CBMs.

**Scalability to massive datasets**   The method for estimating the impact on the risk of adding a new concept can become computationally expensive for large-scale datasets. For such cases, numerical methods for solving linear equations, such as EigenPro [1], or other risk estimation methods, should be considered to improve efficiency without compromising accuracy.

**Inheritance of the identifier's bias**   CBMs may inherit biases present in the training data or external knowledge sources, as discussed in Appendix G. More research is needed to mitigate potential biases and guarantee equitable outcomes.

# F  Extended Future Work

**Transference of Concept Properties**   Future work could further explore the CBM assumption that beneficial inductive biases for humans extend to specific output models. This could involve investigating how different model architectures and training paradigms impact the transferability of these biases.

**Operationalizing Concepts**   Considering concepts through the lens of our characterization provides a principled approach to operationalizing them. Methods to generate concept sets based on their ability to decrease entropy or enhance separability could be developed for unsupervised settings, potentially leveraging statistical measures and techniques like the finite neural tangent kernel for deeper analysis.

**Derivation from Models**   Concepts can be extracted from trained models during a 'model sleeping' phase, where latent representations are analyzed and ranked according to their efficacy in reducing complexity and enhancing task performance. This approach could provide insight into the most informative concepts learned by the model.

**Meta-Learning and Transfer Learning**   Conventional meta-learning and transfer learning methods often integrate datasets with shared input features and targets. A concept-based representation could facilitate data transfer across diverse input spaces and enhance the interpretability, generalizability, and robustness of the model to data shifts. Leveraging large-language models (LLMs) to identify

relevant concepts for a target task from a pool of knowledge datasets offers a novel pathway for knowledge transfer.

**Hierarchical Concept Structures**   Introducing a hierarchical structure of concepts as initially explored by [31, 36] could provide a more nuanced understanding and representation of the data, mirroring human cognitive processes. Although challenging to achieve with foundational models, this approach could be explored by recursively distilling the most valuable concepts from successive layers, enhancing the model's ability to capture complex relationships.

**Broader Implications and Ethical Considerations**   Future research should also consider the broader implications and ethical considerations of using concept-based models. This includes addressing potential biases, ensuring fairness, and evaluating the impact of these models in various real-world applications.

## G   Broader Impacts

Although our main contributions can be considered foundational research and raise no immediate fairness concerns, discussing the presence of biases in concept identification models for practical use cases is relevant. These models have been shown to inherit and propagate societal biases [3], potentially leading to unfair or discriminatory representations that will affect the resulting concept representations.

Several strategies exist to mitigate these risks. First, we should ensure that we use models whose training datasets are diverse and representative, thoroughly evaluated for bias, and employ techniques to detect and correct biases. In particular, we can focus on models that stress transparency by providing comprehensive documentation of datasets and detailing bias detection and correction efforts.

However, provided that we used a model with solid guarantees of fairness, our conclusions suggest that concept representations arising from such models can help reduce the bias inherited by trained models on potentially biased datasets.

