# OpenReview forum: "A theoretical design of concept sets: improving the predictability of concept bottleneck models"
_NeurIPS.cc/2024/Conference — NeurIPS 2024 poster_

### Official Review · Reviewer_QdnR · 2024-07-12

**Soundness:** 3
**Presentation:** 2
**Contribution:** 3
**Rating:** 7
**Confidence:** 1

**Summary:**

This paper provides a theoretical analysis of properties of concept sets in CLIP-based CBMs. Specifically, the authors focus on the effect of concept sets on the empirical performance of CBMs in the low-resource regime and under distribution shifts.  Towards this, the authors identify two characteristics of concept sets that help improve CBM performance: the size of CBMs and the degree of misspecification. The authors validate their findings on two datasets: CIFAR-10 (for evaluating image classifier performance) and MetaShift (for evaluating distribution shift).

**Strengths:**

__Significance__: A lot of recent work has provided empirical efficacy of using CLIP-based interpretable models for few shot classification [1-4]. In practice, the concepts that achieve a high CLIP activation are often nonsensical. A theoretical understanding of the underlying mechanisms behind CLIP based CBMs are extremely desirable and relevant to the NeurIPS community.

`[1]`: https://arxiv.org/abs/2211.11158

`[2]`: https://arxiv.org/abs/2308.03685

`[3]`: https://arxiv.org/abs/2404.09941

`[4]`: https://arxiv.org/abs/2210.07183

**Weaknesses:**

_(This work is outside my area of expertise; I have slight empirical familiarity with CBMs, but I cannot comment on the mathematical correctness of the proofs. Most of my comments are in the questions section)._

__Lacking experimental evaluation__: We have much better datasets than CIFAR-10 to measure the efficacy of an image classifier. I recommend the authors look at LaBO's `[1]` experimental evaluation section for a more comprehensive set of experiments.

Overall, I'm recommending a __Weak Accept__. Theoretical research in the properties of CLIP based CBMs is deeply desired and relevant to the NeurIPS community. I haven't given a higher score because it's hard to make informed opinions based on the current evaluation setup.

**Questions:**

Questions while reading the paper:
- L23: I think CBMs can be motivated with a better example. Here is another example: A “Tesla model X” did not exist when CLIP was trained, yet we can make a reasonably accurate classifier with a concept bottleneck: {SUV, T logo, gull wing doors, (lack of) exhaust pipe}.
- L99: What is P here?
- L106: (Definition 1) What is a non-trivial instantiation of a set that is rejected by this definition (ie: a set which is not a concept set)?
- L154: I understand that the theoretical justification necessitates linearity, but empirical studies suggest that the final CLIP activation is based on a hierarchical composition of neuron activations `[5]`. Can such relationships be captured by the linear model?
-  L248: I agree that removing redundant concepts is important. However, in practice, redundant concepts seem to help improve performance (`[4]` identifies almost 3000 concepts for some datasets).
- L267: I recommend the authors try out more datasets than just CIFAR10 `[6]`. I think following the experimental section for LaBO `[1]`, one of the paper's cited, would be a good idea here.I recommend testing in domain generalization on CINIC-10 `[7]` as well.
-  L275: The appendix mentions using ViT B-32 while this section mentions using ViT L-14. The provided code shows references to both ViT B-32 (for CLIP embedding generation) and ViT L-14 (for querying DINO features; which I'm assuming isn't part of this paper). A clarification of which model was used in which context would be beneficial!


`[5]`: https://distill.pub/2021/multimodal-neurons/

`[6]`: https://arxiv.org/pdf/1806.00451

`[7]`: https://arxiv.org/abs/1810.03505


-----
Increasing my score to __Accept__ after discussions with authors.

**Limitations:**

The authors have adequately addressed limitations in the Appendix. I appreciate the inclusion of a broader impacts section!

---

> ### Author Rebuttal · Authors · 2024-08-07
>
> Thank you for your in-depth review and for highlighting this theoretical work's high relevance to the NeurIPS community.
> ## New experiments
> The purpose of the experiments in our paper is to illustrate the theoretical insights, and we believe the current experiments serve that purpose well. However, to validate our insights further, and as per your comment, we have run experiments on additional datasets as suggested, including CINIC-10 for domain generalization. We refer to the 'Global Response' for the new results.
> ## Questions
> We emphasize our gratitude for the careful review shown by the questions posed, notably since each led to valuable improvements. Next, we indicate the resulting improvements to the paper.
> ### Clarity
> * We address the ambiguity regarding the meaning of $P$ in L99 with the following modification: "[...] $\\mathcal{D}=\\{(x_i, y_i)\\sim_{iid} \\mathbb{P}(\\mathcal{X}, \\mathcal{Y})\\}$ is a dataset drawn from the data-generating process $\\mathbb{P}(\\mathcal{X},\\mathcal{Y})$ [...]".
> * Thanks for flagging the model typo. We use ViT B-32; we have updated it in the text in L275-276.
> ### Linearity
> Indeed, CLIP embeddings of images $CLIP:\\mathcal{X}\\rightarrow \\bar{\\mathcal{X}}$ are non-linear and result from a hierarchical composition of neurons. Our analysis does not assume linearity of such a mapping, but rather linearity of the mapping from the CLIP embedding $\\bar{\\mathcal{X}}$ to the concept representation, which is consistent with the CLIP objective [2] as stated in L150 ("where it assumed that concepts can be related to directions"), as well as linearity
> of the output layers, a standard practice for CBMs [1] and when using CLIP embeddings [2].
>
> We understand our usage of $\\mathcal{X}$ to denote inputs from this embedding space might confuse readers, so we will denote the CLIP embedding space as $\\bar{\\mathcal{X}}$ throughout the paper and modify L149 to "To that end, we consider the input space $\\bar{\\mathcal{X}}$ as the joint embedding space of inputs and concepts from a foundational model [...]".
> ### Redundant concepts
> We agree that a more nuanced discussion would benefit L248.
>
> As per Theorem 1, having a concept that does not decrease the misspecification $\\varepsilon$, which can happen either if it is spanned by the other concepts or orthogonal to the target function, will be detrimental to the risk since it increases the first term of the risk (see L177-178). This is what we mean by a *redundant concept*.
>
> However, there will be many cases where the given concept will not be entirely redundant (i.e., only slightly affect $\\varepsilon$), and in these cases, the choice of keeping or discarding it will depend on the number of samples we have. Both our theoretical and empirical insights (see L281-293) conclude that in lower sample regimes, we should be more aggressive at discarding slightly *redundant concepts*, and as $n$ grows, we will benefit more from preserving them, as you point out.
>
> We appreciate that you flagged the importance of providing a more comprehensive discussion of when to "remove *redundant concepts*" in L248 and connecting it to the theoretical insights. We have correspondingly included the above discussion as a footnote.
> ### Examples
> #### Motivation for CBMs
> Your provided example nicely illustrates how CBMs can exploit the composition of concepts to perform tasks beyond CLIP's training distribution. We have correspondingly added a version of it in line 24. We believe it best to keep the presentation of CBMs in the first paragraph agnostic to the identification method (CLIP) for generality, so we slightly modified it to the following phrasing without discussing CLIP: "Indeed, despite never having seen a Tesla Model X, someone could reasonably classify them provided they can identify an SUV, a T logo, gull-wing doors, and the absence of exhaust pipes."
> #### Non-Concepts
> Let us start with a specific example and then move into a more formal characterization.
> * We humans do not develop some concepts for fields outside our areas of interest (our interest is informative of our underlying task $\\mathcal{T}$) despite experiencing occurrences of such concepts. Consider, for example, filmography concepts such as "Jump cut," where two sequential shots of the same subject are taken from camera positions that vary only slightly (or a "Dutch tilt"). According to Def. 1, for people not interested in filmography, these characteristic functions would not be included in the concept set because they add negligible expressiveness to their current concept set, i.e., $N_0\\rightarrow \\infty$ without significantly contributing to the model-aware inductive bias, so $N_1 < N_0$, failing the definition.
> * More abstractly, a characteristic function $c$ that does not provide additional valuable information on top of the existing set $\\mathcal{C}$ for $\\mathcal{T}$ will have low expressiveness (i.e., large $N_0$) without increasing $N_1$. This can occur when targets are independent of $c$ conditioned on $\\mathcal{C}$: e.g., concepts irrelevant to the task (e.g., tinted glass when predicting a Model X) or completely redundant with $\\mathcal{C}$. It can also happen when the output model cannot capture the correlation, e.g., $c$ might help cluster the input distribution into two concentrical spheres, but a linear model would not benefit from this. Further still, even if a characteristic function $c$ provides information that can be captured, if the inductive bias of the method disprefers the optimal hypothesis given this representation, once again, $N_0$ will be large.
>
> We believe this discussion can be valuable to some readers, so we incorporate it in a new appendix and reference it in L109.
> ## References
> [1] P. W. Koh et al., "Concept Bottleneck Models".
>
> [2] A. Radford et al., "Learning Transferable Visual Models From Natural Language Supervision".

---

> ### Comment · Reviewer_QdnR · 2024-08-07
>
> Thank you for the response! I think the additional studies address my main concern regarding evaluation setup. I'm increasing my score to recommend Acceptance.

---

> > ### Author Response · Authors · 2024-08-08
> >
> > Dear Reviewer,
> >
> > Once more, thank you very much for your thoughtful and constructive review. We are pleased your main concerns were addressed and grateful for your increased acceptance score.

---

### Official Review · Reviewer_sfwM · 2024-07-12

**Soundness:** 3
**Presentation:** 2
**Contribution:** 3
**Rating:** 6
**Confidence:** 3

**Summary:**

This paper presents theoretical contributions related to CBM, which delves into the impact that the choice of concept set has on CBM performance. It identifies advantageous conditions for CBMs, offering an orthogonal and meaningful perspective compared to most other works on CBMs.

**Strengths:**

- Overall, the theoretical findings are insightful. The authors demonstrate the situations in which the performance of CBMs can be enhanced by well-choosing the concept sets.
- The framework of the theories is clear. The assumptions are appropriate and not overly restrictive. The definitions are comprehensive and primarily clarify the expressiveness and model-aware inductive-bias, which are two key properties affecting the performance.
- The empirical results show that the CBMs surpass the foundational embedding models. These results also provide a unique perspective compared to existing works.

**Weaknesses:**

- The multi-step approach introduced in Sec.5.2 seems to be difficult and brings uncertainty from LLMs, which to some extent limits the practicality and generality.
- It would be better if more baselines and datasets were provided in the experiment part. Additionally, due to the costly training procedure, this paper only incorporates the concept representations from foundational multimodal models. This is reasonable but complicates comparisons with other CBMs.

**Questions:**

None.

**Limitations:**

Yes.

---

> ### Author Rebuttal · Authors · 2024-08-07
>
> We thank you for your thoughtful evaluation and appreciate your recognition of our theory's insightfulness and clarity, as well as our results' unique perspective.
> ### The use of LLMs for concept generation
> The core objective of our method and experiments is to illustrate and validate our theoretical insights, which are agnostic to how concepts are sampled. In fact, we note that the first mention of an LLM as a generator happens in L229 in section 5.2. Consequently, using an LLM for sampling concepts in our experiments does not affect the practicality or generality of our theoretical insights. We employ LLMs to reduce the required human expertise and flexibly supply task-specific concepts, decreasing the barrier to adopting CBMs for new tasks.
>
> As mentioned, our theory encompasses concept sets that are not dependent on LLMs. Options include human-generated concepts, concept graphs like ConceptNet, or leveraging correlations within text corpora like Wikipedia. Even hybrid approaches could be considered, with the goal of enhancing LLM generation by conditioning their context on external knowledge sources like ConceptNet or Wikipedia, as mentioned in L229-232.
>
> Orthogonally, we also note that the method in section 5.2 can be regarded as a 'rejection-sampling loop' that partially palliates the practical limitations of relying on an LLM. This loop controls the quality of LLM generation through steps 2 and 3 by rejecting concepts that do not align with our theoretical criteria, thereby encouraging only high-quality concepts to be incorporated into the concept set.
>
> To convey more effectively that this choice does not limit the practicality and generality of our theory and insights, we will include a version of the first two paragraphs above following L232 and a version of the third preceding L254.
> ### New experiments
> The purpose of the experiments in our paper is to illustrate the theoretical insights, and we believe the current experiments serve that purpose well. However, to further validate our insights and in agreement with your comment, we have run experiments on additional datasets and a new baseline. We refer to the 'Global Response' for the new results.
>
> ---
> Once again, we express our gratitude for your comments. We hope that the new experiments and the discussion we incorporate about how our theory is agnostic of the generation method have addressed all your comments, and we look forward to your continued engagement with our research.

---

> > ### Comment · Reviewer_sfwM · 2024-08-09
> >
> > Thanks for the authors' response. Most of my concerns have been addressed and I'm glad to see the authors had supplemented some empirically results to better show the effectiveness. By the way, I have also tried to run Label-free CBM with VIT-B/32 backbone on CIFAR10 and the result is about 86.5%, a bit higher than that in Figure 2 in the rebuttal PDF (a typo here, Figure 2 should be Figure 2b I think, and the left one should be Figure 2a not 2c). Could this discrepancy be due to differences in training size?
> >
> > I have read the comments and responses from the other reviewers and currently have no further questions. However, I found the responses difficult to read due to the extensive use of LaTeX formatting. It would be helpful to use markdown formatting in the rebuttal period for improved readability.

---

> > > ### Author Response · Authors · 2024-08-09
> > >
> > > Dear Reviewer,
> > >
> > > Thank you for your continued engagement and the thoughtful follow-up. We are pleased that the additional experimental results addressed most of your concerns.
> > >
> > > Regarding the performance discrepancy you observed with Label-free CBM using the ViT-B/32 backbone on CIFAR-10, this is indeed primarily due to differences in training size. As discussed in our global response, we also made adjustments to the concept set in the fifth code block of the `GPT_conceptset_processor.ipynb` file from the available Label-Free CBM repository, specifically modifying the concept set sampler $c_i \\sim L(c | C_T)$ to ensure a fair comparison between Label-free CBM and our method. Nonetheless, this modification led to a slight improvement in the performance of Label-free CBM on CIFAR-10, where we achieved an accuracy of $86.79\\%$, which aligns closely with your results.
> > >
> > > To clarify and avoid any confusion, we will add a final data point to Figure 2 in the attached PDF, showing the performance on the full CIFAR-10 dataset. Here, our method achieves an accuracy of $92.23\\%$, thus maintaining an advantage over the baseline comparable to previous data points.
> > >
> > > You are correct about the typo in the figure labels; the left figure in the second row should be labeled "Figure 1c." We have corrected this in the revised document.
> > >
> > > Regarding your feedback on using LaTeX formatting, we apologize for any readability issues you encountered. On our end, the equations render correctly, so we are not entirely sure what specific formatting challenges you experienced. However, we fully appreciate the importance of clarity and are more than willing to reformat the content using markdown or any other format you prefer. If you could provide more details or specify your preferred format, we would happily make the necessary adjustments to ensure the content is as accessible as possible.
> > >
> > > We hope that these clarifications address all your remaining concerns. We also hope that the improvements and additional results presented across the rebuttal will prompt you to consider raising your evaluation of our work.
> > >
> > > Thank you again for your valuable feedback and time.

---

> > > > ### Comment · Reviewer_sfwM · 2024-08-09
> > > >
> > > > All the LaTeX formulations are correct this time, even though I didn't make any changes...sorry for the confusion. Thank you for your reply and I will take your feedback into account.

---

### Official Review · Reviewer_7Ykd · 2024-07-20

**Soundness:** 3
**Presentation:** 2
**Contribution:** 3
**Rating:** 5
**Confidence:** 4

**Summary:**

This paper addresses an important research question in CBM — understanding the properties of concept sets and their connection to the performance of CBMs.
A theoretical framework for concept sets is proposed, focusing on two desiderata for concepts: expressiveness and model-aware inductive bias.
Their theoretical analysis covers under what conditions the concept representation-based predictions outperform raw feature representation-based predictions.
Empirical evaluation with two datasets confirms their theoretical insights that well-chosen concept sets improve sample efficiency and robustness under distribution shifts.

**Strengths:**

Studying the properties of concept sets and how to effectively find a good concept set satisfying the properties, ultimately improving the utility of CBMs, is a very interesting and important research question.
The two proposed properties of concept expressiveness and model-aware inductive-bias are intuitive and well-defined.
Especially, their observation on connecting them to improved data efficiency is interesting.
The proposed theoretical framework naturally reveals the regimes on which it is undesirable to have a concept bottleneck and leads to a principled solution based on the theoretical insights.

**Weaknesses:**

Notations are very loosely defined:
* Notations in introduction are used without being defined; for instance, $f, g, \theta, \mathcal{H}$.
* in line 104, the definition of function f as a concatenation of a mapping $\hat{g}_m$ and a set $[c_1, \dots, c_m]$ is awkward.
* what is $d$ in Theorem 1?
* in line 154, please clarify that $x$ is not a raw input (e.g., image), but a feature representation output from a backbone foundation model.

Limited experiments:
* Evaluated only with two datasets.
* No performance comparison with other CBM methods, given the same number of concepts and training set size.
* No demonstration and anlysis on the actual results of generated concept sets

Please increase the fonts in the figures for visibility.

**Questions:**

* in lines 135-136, `This flexibility means we can replace the underlying ... while keeping the output model unchanged.` Could you please elaborate on this? I reckon the output model should also be changed when the coupled concept identification model is changed.
* in lines 137-139, `The concept set serves as a stable communication interface ... potentially harmonizing datasets and modalities`. Please extend these statements and explain the broader potential impact of this perspective using concept representations.
* how do you determine the thresholds in concept generation? (i.e., $\Omega_m$, $R_m$).

**Limitations:**

Limitations are described in Appendix C.

---

> ### Author Rebuttal · Authors · 2024-08-07
>
> Thank you for your thoughtful review. We are glad you find the research question important and the insights interesting.
> ## Notation
> We agree with all the suggestions and have incorporated them. Specifically:
> * Changes in the introduction:
> 	* We remove $\\theta$.
> 	* L22: we introduce $\\hat{g}$ with "[...] a CBM first identifies key concepts such as the presence of a semaphore or a pedestrian crossing the road ($C: \\mathcal{X}\\rightarrow\\mathcal{C}$), and learns $\\hat{g}:\\mathcal{C}\\rightarrow\\ \\mathcal{Y}$ to act based on them $(\\hat{g}\\circ C)(x)$".
> 	* L33: we define $\\mathcal{H},\\mathcal{H_C}$: "[...] the degree of misspecification $\\varepsilon$ as key drivers of the CBM performance, where $\\varepsilon=|| \\text{arg min}\_{f\\in\\mathcal{H}} \\ell(f) - \\text{arg min}\_{g\\in\\mathcal{H_C}} \\ell(g\\circ C)||$, with $\\mathcal{H}\\subseteq \\{f:\\mathcal{X}\\rightarrow\\mathcal{C}\\}, \\mathcal{H_C}\\subseteq \\{g:\\mathcal{C}\\rightarrow\\mathcal{Y}\\}$ the hypotheses spaces."
> 	* L48: the additions "concept-based models ($\\mathcal{R}(\\hat{g}\\circ C)$, $\\hat{g}\\in \\text{arg min}\_{g\\in\\mathcal{H_C}} \\ell(g\\circ C)$)" and "baseline counterpart ($\\mathcal{R}(\\hat{f}), \\hat{f}\\in\\text{arg min}\_{f\\in\\mathcal{H}} \\ell(f)$)" further clarify $\\hat{g}$ and $\\hat{f}$.
> * L104: we thank you for pointing out this detail since there was a technical condition missing: either (1) if concept sets are unordered, then $\\mathcal{H_C}$ must be a complete set of maps under permutation symmetry, and the learning method invariant under feature permutations, which holds for the output methods considered in the paper; or (2) $\\mathcal{C}$ is a concept sequence, not a set. In the first case, the composition is well-defined since it is equivalent across concept reorderings; in the second, $\\cup$ should be substituted by $\\times$. We make this explicit in L101 by adding, "Consider the output hypothesis spaces $\\mathcal{H_C}\\subseteq\\{g:\\mathcal{C}\\rightarrow \\mathcal{Y}\\}$ complete under permutation symmetry and the learning algorithm invariant under feature permutations. We refer to Appendix B for a technical discussion of this assumption.". We correspondingly added Appendix B with an extension of the above discussion, including the full alternative definition and what methods satisfy assumption (1).
> * We update Thm 1's statement to "Under the above setting and assuming that $\\mathbb{P}$ is an isotropic distribution on the $d-$dimensional input" to introduce $d$ unambiguously.
> * To avoid confusion, we will denote the CLIP embedding space as $\\bar{\\mathcal{X}}$, and we modify L149 to "[...] input space $\\bar{\\mathcal{X}}$ as the joint embedding space of inputs and concepts from a foundational model [...]".
> * We increased the figures' fonts. This is shown in the new figures; previous ones have been modified correspondingly.
> ## New experiments
> The experiments in our paper aim to illustrate the theoretical insights, and we believe the current experiments serve that purpose well. However, in agreement with your comment, we have run experiments on additional datasets and a new baseline to further validate our insights. We refer to the "Global Response" for details. We also included throughout appendices C.4-6 filtered concept sets from our experiments.
> ## Concepts as a stable interface
> > This flexibility means we can replace the underlying concept identification model with a more advanced version in the future while keeping the output model unchanged. The concept set serves as a stable communication interface with fixed dimensionality and functionality. Additionally, concepts can act as common representations across different feature sets, potentially harmonizing datasets and modalities.
>
> We welcome the opportunity to expand on these observations, which cover exciting properties that could lead to impactful future works.
>
> Suppose we aim to learn $f^*:\\mathcal{X}\\rightarrow \\mathcal{Y}$, and let us consider a CBM counterpart, with the concept representation $\\mathfrak{C}^*:\\mathcal{X}\\rightarrow \\mathcal{R}\_{\\mathcal{C}}$, followed by a mapping $f\_\\mathcal{C}^*:\\mathcal{R}\_{\\mathcal{C}}\\rightarrow \\mathcal{Y}$ from concepts to targets. By keeping the concept set $\\mathcal{C}$ fixed (and thus the representation $\\mathcal{R}\_{\\mathcal{C}}$), we can replace an estimator of the concept mapping $\\hat{\\mathfrak{C}}$ by another $\\tilde{\\mathfrak{C}}$ while the probe $\\hat{f}\_\\mathcal{C}$ remains effective, only affected by a distribution shift. E.g., if we train the probe on a CLIP-based concept representation and migrate to a new CLIP2, our probe will remain applicable to the updated and more precise embedding. However, a probe cannot be swapped from the CLIP raw representations to the ones from CLIP2 since they will have different metrics or even dimensionality. That is, the concept representation $\\mathcal{R}\_{\\mathcal{C}}$ is a stable interface with fixed dimensionality and meaning.
>
> Furthermore, suppose we have datasets with different features or modalities $\\mathcal{X}_1, \\dots,\\mathcal{X}_k$ (e.g., images from different sensors or angles). In that case, we can use a common concept set $\\mathcal{R}\_{\\mathcal{C}}$ which has fixed semantic meaning, and project all inputs there $\\mathfrak{C}^*_i:\\mathcal{X}_i\\rightarrow \\mathcal{R}\_{\\mathcal{C}}$, thereby harmonizing them.
>
> Some readers might be interested in delving deeper into these ideas, so we include a version of the above text in the appendix and refer to it in L139.
> ## Thresholds
> We keep a concept if it improves the estimated risk, i.e., $R_m=0$, or $\\hat{\\mathcal{R}}(\\hat{f}\_{C_i}) - \\hat{\\mathcal{R}}(\\hat{f}\_{C_{i-1}})>0$, and we set the entropy threshold proportionally to the number of buckets $n_b$ used to estimate $\\Omega$ to ensure not all samples fall into the same bucket (i.e. $\\Omega_m=\\frac{1}{n_b}$). We will specify this as a footnote in L253.

---

> > ### Comment · Area_Chair_Hv2L · 2024-08-13
> >
> > Dear Reviewer,
> >
> > I would appreciate if you could comment on the author's rebuttal, in light of the upcoming deadline.
> >
> > Thank you,
> > Your AC

---

### Author Rebuttal · Authors · 2024-08-07

We sincerely thank all reviewers for their thoughtful and insightful feedback. We appreciate their recognition of the relevance and importance of our research on understanding the properties of concept sets and their impact on Concept Bottleneck Models (CBMs). We value their appreciation of the provided theoretical insights, and we are committed to addressing their comments and questions to improve our work based on their feedback.

While we provide individual responses to address specific questions, we would like to share globally the **new results** we obtained during the rebuttal process in response to the reviewers' comments. The primary goal of the experiments in our paper is to illustrate our theoretical insights, and we believe the current experiments effectively fulfill this purpose. However, based on the reviewers' suggestions for a more comprehensive set of experiments encompassing additional datasets and baselines, we have included a new out-of-distribution **(OOD) generalization experiment on CINIC-10** [1] and three new experiments on **sample efficiency on CUB-200-2011** [4], **Food 101** [3], and the **Describable Textures Dataset** [2]. Furthermore, we run an experiment on CIFAR-10 **comparing our method with Label-free Concept Bottleneck Models** [5], a meaningful baseline from the literature. The results, including four plots and a table, are provided in the supplementary PDF. We discuss the results and experimental setup in detail in a new appendix. These additions are referenced in the main text at L274, where we discuss the datasets used, as well as at L293 and L326, to point the reader to the additional experiments.

Next, we include an adapted version of the appendix to discuss the new experiments.
## New Data-efficiency Experiments
Responding to the reviewers' call for more comprehensive evaluations, we conducted three new experiments focusing on the data efficiency of CBMs. These experiments compare the performance of a linear output layer on top of the CLIP representation against our concept-bottleneck counterpart across various sample sizes, averaged over five seeds. The datasets used include CUB-200-2011, Food 101, and the Describable Textures Dataset.

**The results reaffirm the insights from theorem 1**. In the small sample regime, the first term of the risk dominates and predicts increased sample efficiency for the CBM compared to the baseline.

However, as the sample size increases, we observe the gap in performance decreases and even reverses in the large sample regime, which is explained by the fact that
> [...] the $\\varepsilon$ coefficient grows with the number of examples $n$

That is, the information loss resulting from the bottleneck, encapsulated in the $\\varepsilon$, becomes dominant and determines the asymptotic gap between the CBM and the baseline.
## New OOD Experiment on CINIC-10
To further validate the robustness of CBMs to distribution shifts, we conducted an OOD generalization experiment on CINIC-10. This experiment used CIFAR-10 images for training, and ImageNet subclasses from the CINIC-10 test set, conforming to a shifted distribution setting.

**Results show that CBMs exhibit superior OOD generalization capabilities**, aligning with our previous results on the 12 datasets from the MetaShift collection.
## New Baseline: Comparison with Label-Free Concept Bottleneck Models
We reiterate that the main contributions from our work lie in the theoretical insights, and our method and experiments are tailored to support these insights. Nonetheless, as per reviewer comments, we have run an additional experiment on CIFAR-10, comparing the performance of our method with label-free CBM [5] as a baseline.

For a fairer comparison of the concept set selection mechanisms, we provide both methods with the same sampling mechanism $c_i \sim L(c | C_T)$ by fixing the language model $L$ and the context we feed it $C_T$. We also use VIT-B/32 for both methods. We then use CLIP and a linear probe, as described in [5], to obtain the baseline. We average the results of training the output probes over five seeds for each training size.

The results show that **our CBM consistently outperforms the baseline**.

---
We hope the additional experiments address the reviewers' concerns; we believe these enhancements strengthen our submission and look forward to further feedback.

## References
[1] L. N. Darlow, E. J. Crowley, A. Antoniou, and A. J. Storkey, "CINIC-10 is not ImageNet or CIFAR-10," Oct. 02, 2018, arXiv: arXiv:1810.03505. doi: 10.48550/arXiv.1810.03505.

[2] M. Cimpoi, S. Maji, I. Kokkinos, S. Mohamed, and A. Vedaldi, "Describing Textures in the Wild," in 2014 IEEE Conference on Computer Vision and Pattern Recognition, Jun. 2014, pp. 3606–3613. doi: 10.1109/CVPR.2014.461.

[3] L. Bossard, M. Guillaumin, and L. Van Gool, "Food-101 – Mining Discriminative Components with Random Forests," in Computer Vision – ECCV 2014, D. Fleet, T. Pajdla, B. Schiele, and T. Tuytelaars, Eds., Cham: Springer International Publishing, 2014, pp. 446–461. doi: 10.1007/978-3-319-10599-4_29.

[4] Wah, C., Branson, S., Welinder, P., Perona, P., and Belongie, S., "The Caltech-UCSD Birds-200-2011 Dataset," California Institute of Technology, CNS-TR-2011-001, 2011.

[5] T. Oikarinen, S. Das, L. M. Nguyen, and T.-W. Weng, "Label-free Concept Bottleneck Models," presented at the The Eleventh International Conference on Learning Representations, Sep. 2022.

---

### Decision · Program_Chairs · 2024-09-25

**Decision:**

Accept (poster)

**Comment:**

After a fruitfuful discussion, the reviewers agreed that studying the properties of concepts sets is a very significant - and so far understudied - research question, and that the paper does a good job at defining and analyzing critical properties for concept sets.  The empirical analysis, including the new results provided in the rebuttal, seem sufficient to confirm the theoretical findings.  All in all, this is a solid and insightful contribution.

I encourage the authors to improve the notation and to integrate all suggestions provided by the reviewers.